# SOFTPLUS ATTENTION WITH RE-WEIGHTING BOOSTS LENGTH EXTRAPOLATION IN LARGE LANGUAGE MODELS

## ABSTRACT

Large language models have achieved remarkable success in recent years, primarily due to the implementation of self-attention mechanisms. However, traditional Softmax attention suffers from numerical instability and reduced performance as the number of inference tokens increases. This paper addresses these issues by proposing a new design principle for attention, viewing it as a two-stage process. We first decompose the Softmax operation into a non-linear positivity transformation and an $l_1$-normalisation step, identifying the latter as essential for maintaining model performance. Our first proposal is to replace the standard exponential function with the more numerically stable Softplus activation and introduce a dynamic scale factor based on invariance entropy, creating a novel attention mechanism that outperforms conventional Softmax attention. Our second proposal is to introduce a re-weighting mechanism that sharpens the attention distribution, amplifying significant weights while diminishing weaker ones. This enables the model to concentrate more effectively on relevant tokens, mitigate the attention sink phenomenon, and fundamentally improves length extrapolation. When combined, these changes ensures numerical stability and dramatically improves length extrapolation, maintaining a nearly constant validation loss at $16\times$ the training length while achieving superior results on challenging long-context retrieval tasks and standard downstream benchmarks.

## 1 INTRODUCTION

Self-attention has been primarily responsible for the recent success of Large Language Models (LLMs). Self-attention enables models to assess the importance of different words within a sentence, capturing complex relationships and dependencies in the data. Its effectiveness is mainly attributed to the Softmax operation, as when the Softmax operation is omitted or replaced with alternative activation functions, model performance tends to decline (Wortsman et al., 2023; Ramapuram et al., 2024; Shen et al., 2023). Although widely used, Softmax self-attention has two main limitations. Firstly, it suffers from numerical instability due to the exponential function ($e^x$), especially when scaling model sizes to trillions of parameters (Qi et al., 2024). Secondly, as the token length increases during inference, the attention scores calculated by self-attention become smoother and lack distinct peaks. This "attention smoothing" hinders the model's ability to establish connections between relevant tokens, thereby crippling the length extrapolation capabilities of transformers (Chiang & Cholak, 2022; Veličković et al., 2024). This issue is compounded by the "attention sink" phenomenon, where a few initial tokens, often regardless of their semantic importance, attract a disproportionate amount of attention, leading to suboptimal performance (Xiao et al., 2023). This paper addresses these problems simultaneously by proposing a new design principle for attention mechanisms that offers improved numerical stability and dramatically better performance at large token lengths. Our core proposal is a new attention architecture consisting of two-stages: a **Normalisation Stage** followed by a **Sharpening Stage**.

To develop a better normalisation stage, we first analysed the standard Softmax function. We deconstructed it into its two functional components: a non-linear positivity transformation ($e^x$) and a subsequent $l_1$-normalisation ($l_1$-normal). Our experiments revealed that the $l_1$-normal is the critical component for maintaining model performance. This insight frees us to redesign the normalisation

process for better stability and performance. We replace the exponential function with the more numerically stable Softplus activation and incorporate a dynamic length scale factor based on invariance entropy. This creates our new normalisation stage, a novel mechanism we call *Length Scaled Softplus Attention* (LSSA), which outperforms the standard attention mechanism not only at the training sequence length but also for longer sequences.

However, a normalisation stage alone does not solve the attention smoothing problem. Therefore, we introduce a re-weighting mechanism that sharpens the attention distribution. Applied after the LSSA normalisation, this mechanism amplifies critical token relationships and suppresses noise using a power transformation. This inherently sharpens attention peaks without needing post-hoc fixes like positional interpolation (Chen et al., 2023; Li et al., 2023), which retroactively stretches embeddings but fails to address the root cause of attention smoothing.

The combination of these two proposed mechanism, LSSA for stable normalisation and re-weighting for sharpening, results in our final model, LSSAR. The effectiveness of this two-stage design is starkly demonstrated in challenging "needle-in-a-haystack" passkey retrieval tasks, where LSSAR succeeds far beyond its training length while standard attention fails completely. Furthermore, LSSAR effectively mitigates the attention sink phenomenon by promoting a more balanced attention distribution. LSSAR maintains a nearly constant validation loss even at $16\times$ the training token length and translates its superior internal metrics into better performance on real-world NLP tasks.

In summary, this paper makes several important contributions:

1. We introduce a novel Softmax-free attention mechanism, LSSA, which incorporates a dynamic length scale factor based on invariance entropy, demonstrating superior performance compared to standard attention mechanisms across a variety of sequence lengths.

2. We propose a novel architectural re-weighting mechanism that fundamentally redesigns how attention scores are computed, inherently sharpening token relevance by amplifying critical weights and suppressing noise.

3. When LSSA is combined with the proposed re-weighting method, the resulting model (LSSAR) not only excels in length extrapolation while ensuring numerical stability, as demonstrated by its success in passkey retrieval tasks where standard attention fails, but also translates into superior performance in real-world applications.

## 2 METHOD

The proposed attention mechanism consists of two-stages. The first involves normalising the raw attention scores to create a stable distribution, and is described in Sect. 2.1 and 2.2. The second re-weights this distribution to sharpen focus and enhance length extrapolation, as described in Sect. 2.3.

### 2.1 SOFTMAX DECOMPOSITION: THE NORMALISATION STAGE

Scaled dot-product attention transforms queries ($\mathbf{Q}$), keys ($\mathbf{K}$), and values ($\mathbf{V}$) into an output. First, attention scores $\mathbf{A}$ are produced via Eq. (1). Here, $\mathbf{Q}, \mathbf{K}, \mathbf{V} \in \mathbb{R}^{L \times d}$, where $L$ is the sequence length and $d$ is the dimensionality.

$$\mathbf{A} = \text{Softmax}\left(\frac{\mathbf{Q}\mathbf{K}^T}{\sqrt{d}} + \mathbf{M}\right) \quad (1)$$

The optional term $\mathbf{M}$ is a mask matrix of shape $L \times L$, which is essential in causal self-attention. The mask $\mathbf{M}$ is constructed with zeros on and below the diagonal and $-\infty$ above it, ensuring the attention mechanism only considers past and present tokens. The attention scores are then used to compute the output as $\mathbf{O} = \mathbf{A}\mathbf{V}$.

The Softmax function is a cornerstone of the attention mechanism. It is often assumed that its non-negative outputs are essential for good performance. However, this assumption is questionable, as replacing Softmax with other non-negative activation functions often leads to a decline in model performance (Wortsman et al., 2023; Ramapuram et al., 2024; Shen et al., 2023).

To investigate this, we conducted experiments to test the necessity of non-negative attention scores. As shown in Tab. 1, inverting the Softmax outputs by multiplying them by negative one resulted in a negligible change in validation loss, suggesting that non-negativity is not a critical property. We also re-centred the attention scores by subtracting the row-wise mean, creating a mix of positive

Table 1: Validation loss values for training the GPT-2 model from scratch with the standard and modified attention scores.

| Operation | Softmax | Inverse | Re-centred | Re-centred & $l_1$-norm |
|---|---|---|---|---|
| Loss | 3.1911 | 3.1915 | 3.2008 | 3.1954 |

and negative values[1]. This led to a minor increase in loss. However, when we subsequently applied $l_1$-normal to these re-centred scores, performance was almost fully restored. These results indicate that the crucial component of Softmax is not its positivity but its normalisation property.

Motivated by this observation, we decompose the Softmax operation into a non-linear positivity transformation, $\phi(\mathbf{x}) = e^{\mathbf{x}}$, followed by an $l_1$-norm:

$$\text{Softmax}(\mathbf{x}) = \frac{e^{\mathbf{x}}}{\sum_j e^{x_j}} = \frac{e^{\mathbf{x}}}{\|e^{\mathbf{x}}\|_1} = \frac{\phi(\mathbf{x})}{\|\phi(\mathbf{x})\|_1} \tag{2}$$

This allows us to generalise the attention mechanism's normalisation stage as shown in Eq. (3). The subscripts $i$ and $j$ represent the row and column indices respectively. $\otimes$ denotes element-wise multiplication. The mask $\mathbf{M}'$ is constructed with ones on and below the diagonal and zeros above it. It is evident that the original attention function (Eq. (1)) is a special case of this general form where $\phi(\mathbf{x}) = e^{\mathbf{x}}$. This decomposition allows for greater flexibility in designing the normalisation stage by exploring alternative positivity functions $\phi$.

$$\mathbf{A} = \phi\left(\frac{\mathbf{Q}\mathbf{K}^T}{\sqrt{d}}\right) \otimes \mathbf{M}'$$

$$\mathbf{A}_i \leftarrow \frac{\mathbf{A}_i}{\|\mathbf{A}_i\|_1} \tag{3}$$

## 2.2 LENGTH SCALED SOFTPLUS ATTENTION

Based on the general form of Eq. (3), we now introduce *Length Scaled Softplus Attention* (LSSA), a novel attention variant designed to enhance the scalability and performance of LLMs. LSSA replaces the exponential function with the *Softplus* activation to introduce non-linearity while maintaining smooth gradients essential for stable training dynamics. Empirical testing of several widely-used activation functions in place of $\phi$ in Eq. (3), revealed that Softplus delivers the best performance (see Tab. 3), justifying its adoption. As descrbed below, LSSA also introduces a novel scaling factor that accounts for both the sequence length and the model's dimensionality, thereby addressing limitations associated with traditional attention methods in handling long sequences during inference.

In contrast to scaled dot-product attention, cosine similarity attention employs the $l2$-norm for each row of $\mathbf{Q}$ and $\mathbf{K}$. This approach has been shown to produce more moderate attention weights, which can enhance performance across various tasks (Henry et al., 2020; Dehghani et al., 2023; Liu et al., 2022). However, in high-dimensional spaces, as the number of dimensions increases, two randomly selected vectors are likely to become orthogonal. This phenomenon causes the elements of the product of normalised $\mathbf{Q}_i/\|\mathbf{Q}_i\|_2$ and $\mathbf{K}_i^T/\|\mathbf{K}_i\|_2^T$ to approach zero, thereby worsening the vanishing gradient problem. Consequently, a scale factor becomes essential for cosine similarity attention, and the factor should be associated with the dimensionality $d$.

Previous work demonstrated that replacing the traditional scaling factor $1/\sqrt{d}$ with $\log L/\sqrt{d}$ in scaled dot-product attention enhances the length extrapolation capabilities of transformers (Chiang & Cholak, 2022). Furthermore, Su (2021) highlighted that the inclusion of the $\log L$ factor aids in maintaining entropy invari-

$$\mathbf{Q}_i \leftarrow \frac{\mathbf{Q}_i}{\|\mathbf{Q}_i\|_2}, \mathbf{K}_i \leftarrow \frac{\mathbf{K}_i}{\|\mathbf{K}_i\|_2}$$

$$\mathbf{A} = \text{Softplus}\left((\log d \log \mathbf{N}) \otimes \mathbf{Q}\mathbf{K}^T\right) \otimes \mathbf{M}' \tag{4}$$

$$\mathbf{A}_i \leftarrow \frac{\mathbf{A}_i}{\|\mathbf{A}_i\|_1}$$

ance with different token length, thereby facilitating better extrapolation to unknown sequence lengths. We extend this concept by introducing a dynamic length scale factor that adapts to the varying number of attended tokens in each row of the attention matrix. Specifically, we set the scaling factor to $\log d \log \mathbf{N}$, where $\mathbf{N}$ is an $L \times L$ matrix where each element in row $i$ is equal to $i$ (the number of tokens attended to in that row). This ensures the attention mechanism remains robust across varying sequence lengths. The formulation of LSSA is mathematically defined in Eq. (4).

---

[1]To prevent training instability due to the absence of non-zero values in the initial rows, the first three rows were left unmodified.

It is important to note that due to hardware limitations, the proposed LSSA was evaluated with a sequence length of $L = 1024$ and dimension $d = 64$. For LLMs trained with longer sequence lengths and higher dimensionalities, the scaling factor may require further adjustment. However, the component $\log \mathbf{N}$ should remain unchanged, as it is crucial to maintain the length extrapolation capability of the model.

### 2.3 ATTENTION RE-WEIGHTING MECHANISM: THE SHARPENING STAGE

While the normalisation stage provides a stable attention distribution, it still activates all input tokens. This is counter-intuitive, as tokens early in a sequence are less likely to be relevant to the current token being processed. This leads to the attention smoothing problem (Chiang & Cholak, 2022; Veličković et al., 2024), where attention scores lack distinct peaks as sequence length increases during inference.

Experiments with ReLU-based attention, shown in Tab. 3, demonstrate a minor performance decrease compared to the original attention mechanism. The observed performance decrement can be attributed to the distribution of positive values within the row attention scores. Specifically, positive values are predominantly concentrated around the diagonal, while other positions remain negative. The ReLU activation function suppresses these negative inputs, resulting in a limited number of activated neurons during training. This sparsity restricts the model's capacity to capture essential relationships across the entire sequence, thereby adversely impacting overall performance.

This phenomenon cannot be adequately addressed by traditional normalisation methods such as LayerNorm (Ba et al., 2016), as LayerNorm does not alter the underlying distribution of row attention scores. One potential solution is to transform all attention score values to the positive domain using functions like $e^x$ or Softplus before applying LayerNorm. As all values become positive through such transformations, employing the $l_1$-norm emerges as a more effective strategy. The $l_1$-norm is advantageous because it is simple to calculate and ensures that the mean of the effective values in each row (the positive values that sum to one) remains constant (equals to $1/i$), thereby maintaining the integrity of the attention distribution. This reframes both standard Softmax and our LSSA as a unified **normalisation stage**. This conceptual separation enables us to introduce our re-weighting mechanism, defined in Eq. (5), as a distinct **Sharpening Stage** applied after the initial normalisation.

Here, the normalised scores $\mathbf{A}$ are first scaled by the token count $\mathbf{N}$ and then shifted by a matrix $\mathbf{O}$ (an offset matrix of ones, with zeros in the first three rows to prevent instability). This centres the distribution around zero. The $\mathrm{ReLU}^p$ function then masks scores below zero and sharpens the remaining positive scores by raising them to the power of a

$$\mathbf{A} \leftarrow \mathrm{ReLU}^p \left( \mathbf{A} \otimes \mathbf{N} - \mathbf{O} \right)$$
$$\mathbf{A}_i \leftarrow \frac{\mathbf{A}_i}{\|\mathbf{A}_i\|_1} \tag{5}$$

hyper-parameter $p$. A final $l_1$-normal ensures the output is a valid probability distribution. The re-weighting mechanism is an additional component of our proposed attention layer, we refer to the combination of LSSA (stage 1) and this re-weighting mechanism (stage 2) as LSSAR in the rest of this paper (see Algorithm 1).

This power operation is key to solving the attention smoothing problem. To demonstrate this let $x_1, x_2, \ldots, x_n$ represent the positive elements in one row of the ReLU output from Eq. (5), with the maximum value denoted as $x_m$. The formula for any input $x_j$ in Eq. (5) is $x_j^p / \sum_{k=1}^n x_k^p$. For any $x_l$ where $x_l < x_m$, the distance between their re-weighted values ($\overline{x}_m, \overline{x}_l$) with respect to $p \to \infty$ is given by:

$$\lim_{p \to \infty} \Delta(\overline{x}_m - \overline{x}_l) = \lim_{p \to \infty} \frac{x_m^p - x_l^p}{\sum_{k=1}^n x_k^p} = \lim_{p \to \infty} \frac{1 - (\frac{x_l}{x_m})^p}{\sum_{k=1}^n (\frac{x_k}{x_m})^p} = 1 \tag{6}$$

This limit demonstrates that as $p$ increases, the distance between the maximum value and any smaller value approaches 1. Specifically, when $p$ is large, the term $(x_l/x_m)^p$ approaches zero for any $x_l < x_m$, causing all non-maximum values to be effectively suppressed. Meanwhile, the denominator becomes dominated by $x_m^p$, resulting in the maximum value approaching 1 while all other values approach 0. This property ensures that the attention mechanism maintains sharp, distinct peaks even with longer sequences, thereby preserving the model's ability to focus on the most relevant tokens.

## 3 EXPERIMENTS

The experiments were conducted using 8 NVIDIA A100 80GB GPUs. We utilised the GPT-2 small architecture (124 million parameters; Radford et al., 2019), replacing the original absolute position embeddings with Rotary Position Embeddings (RoPE; Su et al., 2024).

All models were trained using a sequence length of 1024 on the fineweb-10B dataset (Penedo et al., 2024), which consists of 10.2 billion training tokens distributed across 18,865 training steps, along with 0.1 billion validation tokens.

---

**Algorithm 1** High-Level Computation of LSSAR

**Require:** $\mathbf{Q} \in \mathbb{R}^{L \times d}, \mathbf{K} \in \mathbb{R}^{L \times d}, \mathbf{V} \in \mathbb{R}^{L \times d}$
**Require:** $\mathbf{M}' \in \{0, 1\}^{L \times L}$
**Require:** $p > 0$
**Ensure:** $\mathbf{O} \in \mathbb{R}^{L \times d}$
 1: Stage 1: Compute the normalised attention using the LSSA mechanism, using Eq. (4).
 2: Stage 2: Compute the final sharpened attention $\mathbf{A}$ by applying the re-weighting mechanism to the normalised attention, using Eq. (5).
 3: $\mathbf{O} = \mathbf{AV}$
 4: **return O**

---

### 3.1 SOFTMAX DECOMPOSITION

As illustrated in Eq. (2), the Softmax operation can be decomposed into a non-linear transformation ($e^x$), followed by the $l_1$-norm. In this section, we examine the importance of each component for LLMs by training the GPT-2 model from scratch with and without each component.

The results are presented in Tab. 2. Comparing the model employing Softmax in Scenario IV with Scenario I, which does not utilise Softmax, reveals that performance does not significantly deteriorate without Softmax. This can be attributed to the masking operation in attention, which elevates the rank of raw attention scores from $d$ to $L$. Given that $L \gg d$, causal masking substantially enhances the model's non-linear capability. In tasks devoid of causal masking, such as transformers used in vision applications, performance degrades markedly (Wortsman et al., 2023). Scenario II shows that using a non-linearity enhances performance. However, it is possible that alternative activation functions may enhance performance beyond that produced by using the exponential function.

Table 2: Validation loss for GPT-2 model with and without the Softmax components: $e^x$ and $l_1$-norm.

| Scenario | I | II | III | IV |
|---|---|---|---|---|
| $e^x$ | | ✓ | | ✓ |
| $l_1$-norm | | | ✓ | ✓ |
| Loss | 3.4138 | 3.247 | 3.3297 | 3.1911 |

Moreover, the $l_1$-norm also contributes to performance enhancement. The attention score matrix constitutes a high-dimensional linear transformation matrix. By normalising each row of the transformation matrix to ensure that the sum of the absolute values equals one, the $l_1$-norm balances the influence of each dimension in the value vectors $\mathbf{V}$. This normalisation functions similarly to a weighted sum, permitting a more equitable contribution from each feature. Consequently, it stabilises the training process and prevents any single dimension from disproportionately impacting the output, ultimately leading to improved model performance.

### 3.2 COMPARISON WITH DIFFERENT ACTIVATION FUNCTIONS

We modify the standard attention mechanism by introducing several novel attention variants for comparative analysis. Specifically, these variants are created by substituting $\phi$ in Eq. (3) with severalalternative activation functions: ReLU (Rumelhart et al., 1986), ReLU$^2$ (So et al., 2021), ReLU6 (Howard et al., 2017), GeLU (Hendrycks & Gimpel, 2016), Sigmoid (Rumelhart et al., 1986), Softplus (Zheng et al., 2015), and Mish (Misra, 2019).

The results presented in Tab. 3 indicate that all activation functions, except for Softplus, result in poorer validation loss values compared to the standard attention mechanism represented by $e^x$. This phenomenon can be attributed to the distribution of positive values in the row attention scores (the

Table 3: Validation loss for various activation functions employed in the attention mechanism. Note that $\phi = e^x$ corresponds to the conventional Softmax attention.

| | ReLU | ReLU$^2$ | ReLU6 | GeLU | Sigmoid | Softplus | Mish | $e^x$ |
|---|---|---|---|---|---|---|---|---|
| Loss | 3.2006 | 3.2494 | 3.2039 | 3.2051 | 3.2000 | **3.1901** | 3.2001 | 3.1911 |

Table 4: Comparison of state-of-the-art Softmax-free attention mechanisms across different sequence lengths. Bold font indicates the best result for each sequence length.

| Attention Mechanism | 1K | 2K | 4K | 8K |
|---|---|---|---|---|
| Softmax | 3.1911 | 4.1662 | 5.4513 | 6.2823 |
| Sigmoid (Ramapuram et al., 2024) | 3.1935 | 7.4554 | 11.8355 | 14.4995 |
| Sigmoid (Ramapuram et al., 2024)($b = -\log \mathbf{N}$) | 3.1930 | 6.5830 | 8.4679 | 9.5939 |
| Sigmoid (Ramapuram et al., 2024)($b = -\log \mathbf{N}, l_1$-norm) | 3.1849 | 4.3470 | 5.5544 | 6.1846 |
| Sigmoid (Tab. 3) | 3.2000 | 4.3811 | 5.8465 | 6.5701 |
| ReLU (Wortsman et al., 2023) | 3.2143 | 6.2662 | 8.4982 | 10.3460 |
| ReLU (Li et al., 2022) | 3.2006 | 4.5192 | 5.6924 | 6.4561 |
| ReLU (Shen et al., 2023) | 3.2155 | 6.5573 | 8.9072 | 10.7266 |
| LSSA | 3.1905 | 4.1301 | 5.2960 | 5.9403 |
| Re-weighted ($p = 3$) | | | | |
| Softmax | 3.1879 | 4.0277 | 5.2842 | 6.2339 |
| Sigmoid (Ramapuram et al., 2024)($b = -\log \mathbf{N}, l_1$-norm) | 3.1841 | 3.7387 | 4.9286 | 5.7450 |
| LSSAR | **3.1782** | 4.2383 | 5.4056 | 6.3007 |
| Re-weighted ($p = 15$) | | | | |
| Softmax | 5.3878 | 5.9491 | 6.5276 | 7.0183 |
| Sigmoid (Ramapuram et al., 2024)($b = -\log \mathbf{N}, l_1$-norm) | 3.2171 | 3.3499 | 3.6108 | 3.8587 |
| LSSAR | 3.1905 | **3.1930** | **3.2291** | **3.3171** |

inputs to these activation functions), which are predominantly concentrated around the diagonal, while other positions remain negative. Consequently, these activation functions, apart from Softplus, tend to suppress negative inputs, leading to a limited number of non-zero attention scores during training. This suppression results in higher loss values than those observed with $e^x$. Nevertheless, the performance differences are relatively minor, suggesting that the positive values play a more critical role in influencing the model's output.

In contrast to $e^x$, the outputs of Softplus exhibit a slower growth rate, resembling a linear function when inputs are positive. This characteristic impedes its ability to emphasise large positive values, and consequently, the resulting attention focuses more on tokens that are further away from the diagonal, which could explain why its validation loss is lower than that of $e^x$. Further supporting this observation, the SquareReLU function clips negative values at zero and squares the positive inputs, thereby amplifying larger positive values even further, resulting in the highest loss value among the tested activation functions.

## 3.3 COMPARISON WITH STATE-OF-THE-ART SOFTMAX-FREE ATTENTION METHODS

We compared the proposed LSSA and LSSAR with leading Softmax-free methods using the same GPT-2-124M model. These attention functions include two that use Sigmoid as proposed in Ramapuram et al. (2024) and in this paper (the Sigmoid attention used in Tab. 3), and three ReLU-based attention methods from (Wortsman et al., 2023; Li et al., 2022; Shen et al., 2023). For Sigmoid attention (Ramapuram et al., 2024), we evaluated two additional variants. The first variant alters the hyperparameter $b$ from $-\log L$ to $-\log \mathbf{N}$. The second variant maintains this modification while subsequently applying the $l_1$-norm to each row. The experimental findings are detailed in Tab. 4.

At an inference sequence length of 1K, all attention variants perform similarly, with only slight differences in validation loss. Specifically, Sigmoid (Ramapuram et al., 2024)($b = -\log \mathbf{N}, l_1$-norm) and LSSA perform slightly better than the standard Softmax attention. As sequence length increases, performance differences become more evident. The standard Softmax attention experiences performance decline with longer sequences but remains relatively stable compared to most other methods.

Sigmoid attention (Ramapuram et al., 2024) shows significant performance decline at longer sequences, with validation loss rising sharply at 8K tokens. The modified Sigmoid (Ramapuram et al., 2024)($b = -\log \mathbf{N}$) shows some improvement, suggesting that the proposed hyperparameter $\log \mathbf{N}$ is more effective than $\log L$, although the model still faces challenges with longer sequences. Incorporating $l_1$-normal in Sigmoid (Ramapuram et al., 2024)($b = -\log \mathbf{N}, l_1$-norm) aligns its performance more closely with standard Softmax attention, underscoring the importance of proper normalisation in attention mechanisms.

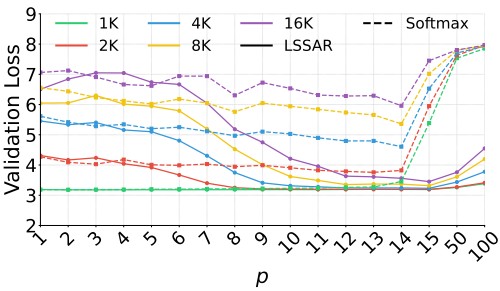

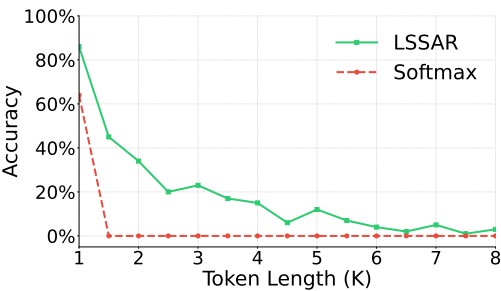

Figure 1: Comparison of LSSA and Softmax attention with varying values of $p$ for the re-weighting mechanism across different sequence lengths.

Figure 2: Passkey retrieval accuracy for LSSAR ($p = 15$) and standard Softmax attention. Accuracy is averaged over 100 trials with the passkey placed at random positions within the sequence.

ReLU-based attention variants (Wortsman et al., 2023; Li et al., 2022; Shen et al., 2023) also experience notable performance decline with longer sequences, though not as drastically as the basic Sigmoid attention. This indicates that while ReLU-based methods may be effective for shorter sequences, they might not be ideal for managing longer-range dependencies.

To demonstrate the effectiveness of the proposed re-weighting mechanism, the attention scores generated by Softmax, Sigmoid (Ramapuram et al., 2024)($b = -\log \mathbf{N}, l_1$-norm), and LSSA were re-weighted with $p = 3$ and $p = 15$. As shown in Tab. 4, the length extrapolation ability provided by the re-weighting can improve the performance of all the tested attention mechanisms. For Softmax, performance was improved slightly using $p = 3$, but was harmed when using $p = 15$. For the other attention mechanisms, $p = 3$ was most effective for sequence lengths of 1K, while $p = 15$ was best for the longer sequences. For $p = 15$ LSSAR exhibits exceptional stability across sequence lengths, maintaining nearly constant validation loss even at 8K tokens. This quantitative robustness is further corroborated by the qualitative attention map visualisations in Appendix A.1 (see Fig. A1), where LSSAR displays a more balanced distribution across tokens with sharper, non-collapsing peaks and markedly reduced attention sink compared to standard Softmax, confirming its enhanced ability to preserve salient long-range dependencies.

### 3.4 ABLATION STUDY FOR RE-WEIGHTING MECHANISM

To gain deeper insights into the proposed re-weighting mechanism, we evaluated it using LSSA and Softmax attention across a range of $p$. As illustrated in Fig. 1, LSSA and Softmax attention exhibit distinct behaviours to changes in the re-weighting parameter $p$. While Softmax attention performs comparably to LSSA at lower $p$ values, it suffers from severe gradient explosion issues as $p$ increases to $p = 15$ and beyond. This instability manifests as a sharp rise in validation loss across all sequence lengths, rendering the model practically unusable at these higher $p$ values.

In contrast, increasing $p$ values generally improve the performance of LSSA across different sequence lengths, with optimal results observed around $p = 15$. The validation loss remains relatively stable as the sequence length increases, underscoring LSSA's strong scalability. This may be because transformers can be viewed as a smoothed cubic spline and Softplus activation is more natural, smooth, and maintains fidelity with spline representations (Lai et al., 2024). In addition, the derivative of the Softplus function is the Sigmoid function, which has an upper bound of one. Hence, LSSA is less susceptible to gradient explosion problems. LSSA maintains stability and avoids gradient explosion even at extreme values of $p$, although we observe a gradual increase in validation loss beyond $p = 15$. This increase occurs because larger $p$ values lead to a sharper attention distribution in each row, causing the model to focus primarily on the maximum value while potentially overlooking other relevant contextual information. This suggests that LSSA can effectively handle large $p$ values without numerical instability, outperforming Softmax attention in this aspect.

Table 5: Zero-shot performance of the models with Softmax attention and LSSAR on downstream tasks. The best scores in each column indicated in bold.

|  | ARC-E | ARC-C | HellaSwag | PIQA | MMLU | SciQ | SummScreen |
|---|---|---|---|---|---|---|---|
| Softmax | 39.77 | **23.72** | 32.42 | 64.09 | 22.97 | 60.6 | 1.682 |
| LSSAR ($p = 15$) | **40.57** | 22.61 | **33.03** | **65.34** | 22.97 | **62.1** | **6.309** |

### 3.5 LONG CONTEXT PASSKEY RETRIEVAL

To offer a more direct and challenging evaluation of length extrapolation, we employed the long-context passkey retrieval task (Mohtashami & Jaggi, 2023). This needle-in-a-haystack test is specifically designed to assess a model's ability to identify a single, crucial, piece of information within a long and distracting context.

We compared LSSAR ($p = 15$) against the standard Softmax attention baseline. The results, shown in Fig. 2, reveal a critical failure in the baseline model. While Softmax attention achieves 64% accuracy within its training length, its performance catastrophically collapses to 0% for all tested lengths of 1.5K tokens and beyond. This demonstrates its inability to overcome attention smoothing, effectively losing the passkey as the context grows. In stark contrast, LSSAR not only achieves a much higher accuracy of 86% at the 1K training length but also demonstrates remarkable robustness in extrapolation. Its accuracy degrades gracefully from 45% at 1.5K tokens, 20% at 4K tokens, and it maintains non-zero accuracy even at 8K tokens. This sustained, non-zero performance provides clear evidence that the re-weighting mechanism successfully sharpens the model's focus, allowing it to pinpoint the relevant information even in sequences far exceeding its training context and overcoming a critical limitation of standard Softmax attention. Additional experiments on passkey retrieval with different model scales and filtered data are provided in Appendix A.2.

### 3.6 DOWNSTREAM EVALUATION

To demonstrate the practical efficacy of LSSAR, we evaluate its zero-shot performance against Softmax attention on six standard benchmarks: ARC (Clark et al., 2018), HellaSwag (Zellers et al., 2019), PIQA (Bisk et al., 2020), MMLU (Hendrycks et al., 2020), SciQ (Welbl et al., 2017), and SummScreen (Chen et al., 2021), using the `lm-evaluation-harness` framework (Gao et al., 2024). For ARC-E, ARC-C, HellaSwag, PIQA, and SciQ, we report normalised accuracy; for MMLU, we use standard accuracy; and for SummScreen, we adopt the ROUGE-1 score to measure summarisation quality.

LSSAR outperforms Softmax attention in five of these tasks, and has performance competitive with that of Softmax attention in the other two (Tab. 5). Notably, LSSAR excels on the long-context summarisation benchmark SummScreen, achieving a nearly fourfold improvement score over Softmax attention. This highlights LSSAR's ability to synthesize information over sequences of up to 2K tokens, aligning with its design for robust length extrapolation. Further downstream evaluation results on different model scales can be found in Appendix A.2.

### 3.7 COMPUTATIONAL ANALYSIS

To evaluate the computational costs of our proposed methods, we conducted a benchmark analysis. The results, detailed in Table 6, compare the performance of GPT-2-124M models with LSSA and LSSAR against the standard Softmax attention baseline. For a fair comparison, all methods were implemented using standard PyTorch functions without leveraging specialised fused CUDA kernels (e.g., FlashAttention). Experiments were conducted on a single NVIDIA A100 GPU using bfloat16 precision with a batch size of 4 and a sequence length of 1024.

The results in Table 6 reveal a critical insight. While LSSAR ($p = 15$) shows a significant increase in *training* memory, its *evaluation* memory footprint is nearly identical to the standard Softmax baseline. This discrepancy is a direct consequence of the PyTorch Autograd engine, which must cache large intermediate tensors for the backward pass of the ReLU$^p$ operation. The near-constant

Table 6: Computational overhead comparison for different attention mechanisms.

| Attention Mechanism | Training | | Evaluation | |
|---|---|---|---|---|
| | Time (ms) | Memory (MB) | Time (ms) | Memory (MB) |
| Standard Attention | 120.48 | 9609.45 | 49.26 | 2324.71 |
| LSSA | 169.92 | 12071.23 | 58.56 | 2325.58 |
| LSSAR ($p = 15$) | 250.32 | 16685.57 | 81.15 | 2325.58 |

evaluation memory proves that the model's intrinsic state is not larger; the overhead is therefore an engineering artefact of the unoptimised backpropagation, not a fundamental algorithmic limitation.

To demonstrate that an optimised implementation is feasible, we analyse LSSAR's theoretical complexity. From a computational standpoint, the complexity of standard attention is dominated by two matrix multiplications, resulting in $O(L^2 d)$ floating-point operations (FLOPs). The additional operations in LSSAR (norms, element-wise functions) are of a lower order ($O(Ld)$ or $O(L^2)$), meaning that the asymptotic computational complexity of LSSAR remains identical to standard attention.

The more critical aspect is memory (I/O) complexity. The primary bottleneck in naive attention is the memory bandwidth required to read and write the large $L \times L$ attention matrix to and from High-Bandwidth Memory (HBM). I/O-aware algorithms like the FlashAttention family (Dao et al., 2022; Dao, 2023; Shah et al., 2024) solve this by computing the output in tiles without ever materialising the full matrix in HBM, reducing memory access complexity from $O(L^2)$ to the optimal $O(Ld)$. Crucially, all additional operations in LSSAR are local (element-wise or row-wise). This locality means they can be applied to a sub-block (tile) of the attention matrix within fast on-chip SRAM. Consequently, these operations can be fused into the main loop of a tiled attention algorithm.

In summary, while our proof-of-concept implementation shows expected overheads, both empirical evidence and theoretical analysis confirm that LSSAR is fundamentally compatible with modern, high-performance attention implementations, with a clear path to achieving computational efficiency on par with standard attention.

## 4 DISCUSSION

Due to limited computing resources LSSAR was evaluated on a small model (GPT-2-124M) with a maximum inference token length of 16K. However, we can infer from Fig. 1 that the validation loss of LSSAR would remain relatively stable for token lengths greater than 16K. Furthermore, in deep neural networks, successive layers progressively refine feature representations through hierarchical transformations, each modulated by learned weights. Analogously, increasing the parameter $p$ in the attention re-weighting mechanism incrementally sharpens the attention distribution. In this context, the extreme values of $p$ in Fig. 1 can be interpreted as effectively increasing the depth of the GPT-2-124M model. Consequently, it is reasonable to expect that LSSAR would maintain comparable stability and performance when scaled to larger models with billions of parameters. Additional experiments (see Appendix A.2) on other scales (45M and 355M) using filtered data, corroborate this expectation.

The implementation of LSSA and LSSAR using PyTorch's built-in functions incurs significant memory and computational costs. For practical applications, it is advisable to develop a highly optimised tensor programme, such as Flashattention-3 (Shah et al., 2024), specifically tailored for these components. However, as this requires a deep understanding of both hardware and CUDA programming, an alternative, less-challenging, approach would be to utilise Mirage (Wu et al., 2024), a tool that automatically searches for suitable CUDA kernels and generates highly optimised tensor programmes.

Additionally, it is recommended to use the T6 model (Zhang et al., 2025) with LSSAR in practical applications. T6 employs tensor decompositions to represent queries, keys, and values compactly, significantly reducing key-value cache memory while achieving better performance than standard transformers. By replacing Softmax attention with LSSAR in T6, the resulting model is expected to not only efficiently process significantly longer sequences under fixed resource constraints but also exhibit improved length extrapolation ability.

## 5 RELATED WORK

**Softmax-free Attention.** Previous research has investigated Softmax-free attention by substituting the Softmax function with the ReLU activation (Li et al., 2022; Shen et al., 2023; Wortsman et al., 2023; Hron et al., 2020; Bai et al., 2024; Fu et al., 2024), SquaredReLU activation (Hua et al., 2022), and Sigmoid activation (Ramapuram et al., 2024), as well as examining purely linear attention (Qin et al., 2022; Tsai et al., 2019; Katharopoulos et al., 2020; Han et al., 2023; Arora et al., 2024; Lu et al., 2021). However, none of these approaches outperform the original Softmax attention. In contrast, the proposed LSSA demonstrates enhanced numerical stability and superior performance across various token lengths.

It is also important to distinguish our work from two other lines of research for long-context modeling. The first improves computational efficiency by approximating the attention matrix, leading to linear attention mechanisms such as Performer (Choromanski et al., 2020). The second line of research proposes entirely new architectures, such as introducing sparse attention patterns as in LongFormer (Beltagy et al., 2020), or replacing the attention paradigm altogether with State Space Models like Mamba (Gu & Dao, 2024). In contrast to these approaches, LSSAR remains within the exact, quadratic-time, attention paradigm. Our contribution is not a new architecture or an approximation, but a direct, drop-in replacement for the standard attention, to fundamentally improve its numerical stability and length extrapolation capabilities within the conventional Transformer framework.

**Attention Re-weighting.** The non-linear re-weighting mechanism introduced by softmax attention ($l_1$-normal) has been shown to concentrate the distribution of attention weights, thereby stabilising the training process (AUEB et al., 2016; Gao & Pavel, 2017; Jang et al., 2016; Qin et al., 2022). Our empirical findings further demonstrate its essential role in maintaining the performance of LLMs. Moreover, we introduce a novel perspective: a non-linear positivity transformation followed by $l_1$-normal. Inspired by the classic normalisation-ReLU structure, the proposed re-weighting mechanism masks less relevant tokens and amplifies the relevant ones, which boosts the length extrapolation ability of underlying models. This provides the deep learning community with a deeper understanding of the attention mechanism within transformers.

**Length Extrapolation.** Positional embeddings (Su et al., 2024; Chen et al., 2023; Chi et al., 2022; Kiyono et al., 2021; Golovneva et al., 2024; He et al., 2024; Huang et al., 2020; Li et al., 2023; Likhomanenko et al., 2021; Liu et al., 2023; Wang et al., 2024; Zheng et al., 2024) play a vital role in transformer architectures by providing essential information about the positions of tokens within a sequence, which is considered a key factor in enhancing length extrapolation (Kazemnejad et al., 2023). Among these embeddings, RoPE is particularly noteworthy; it forms the foundation of many modern LLMs, including GPT-4 (Achiam et al., 2023), Llama3 (Dubey et al., 2024), Deepseek-v3 (Liu et al., 2024), DeepSeek-R1 (DeepSeek-AI et al., 2025) and Qwen3 (Yang et al., 2025). Unlike RoPE-based extrapolation techniques that compensate for sequence length limitations through positional embedding adjustments (Chen et al., 2023; Li et al., 2023; kaiokendev, 2023; bloc97, 2023b;a; emozilla, 2023), LSSAR redefines the core attention mechanism itself. The re-weighting operation is not an auxiliary technique but a structural enhancement to the attention computation, ensuring sharp peaks and stable gradients by design. This indicates that a well-designed attention mechanism like LSSAR is essential for RoPE-based LLMs to enhance their length extrapolation, suggesting that LSSAR may serve as a general strategy to bolster the length extrapolation abilities of most contemporary LLMs.

## 6 CONCLUSION

This paper introduced two novel improvements to attention mechanisms. The first normalises scores using LSSA, a mechanism built on the Softplus function and a dynamic length scale factor, which outperforms standard Softmax. The second applies a unique re-weighting mechanism that sharpens the attention distribution. The combined model, LSSAR, demonstrates a remarkable ability to extrapolate to longer sequences while maintaining numerical stability and mitigating the attention sink phenomenon. Downstream evaluations confirm that these architectural improvements translate into superior performance on a range of tasks. We believe LSSAR offers a robust and effective path forward for future transformer architectures.

# 7 ETHICS STATEMENT

We adhere to the ICLR Code of Ethics. This work involved no human subjects and required no IRB approval. We used publicly available datasets (e.g., FineWeb-10B) under their respective licenses, without processing personally identifiable information. We acknowledge potential societal biases in web-scale data and the risk of misuse of our methods; we will release code with safe-use documentation to encourage ethical compliance. The authors declare no conflicts of interest.

# 8 REPRODUCIBILITY STATEMENT

The methods and experimental setups are described in detail in Section 2 and Section 3, respectively, to ensure the reproducibility of our results. The source code for our models and experiments will be made publicly available upon publication of this work.

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

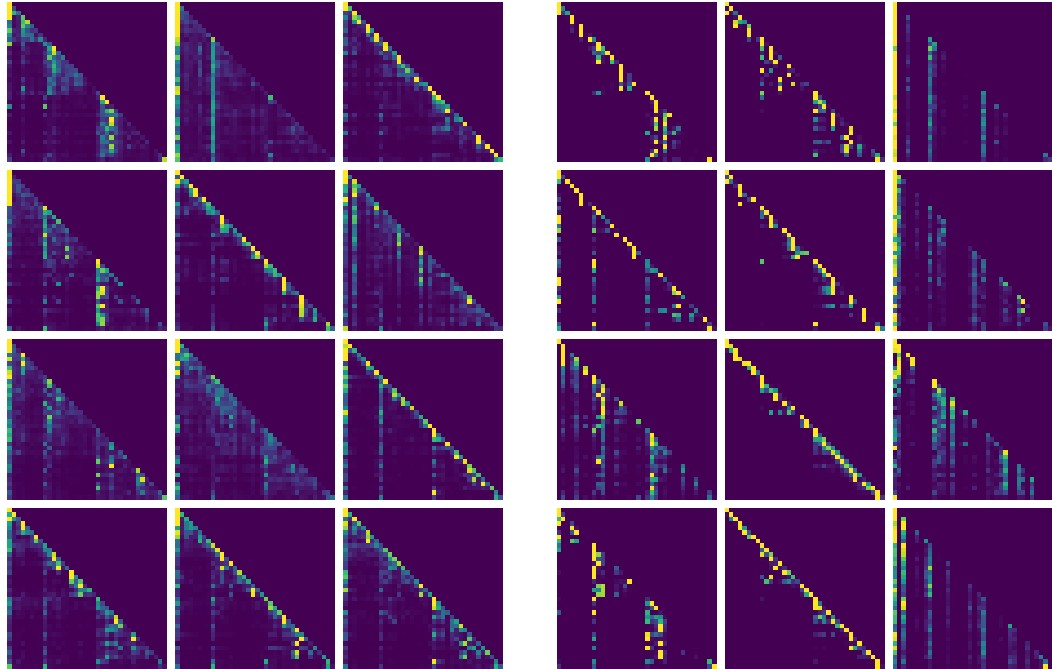

Figure A1: Comparison of attention maps from the last layer of GPT-2-124M, showing standard Softmax attention (left) versus LSSAR with $p = 15$ (right). Each panel displays the 12 attention heads in a 4x3 grid. For visualisation purposes, attention scores are clamped to the range [0, 0.5]. The input text is: *Working from home can be great most days. I enjoy the flexibility and not having to commute in traffic. But sometimes I miss the office interactions with my coworkers and team meetings.*

# A APPENDIX

## A.1 VISUALISATION OF ATTENTION SCORES

This section provides a visual comparison of attention maps generated by standard Softmax attention and our proposed LSSAR mechanism. The attention maps are extracted from the final layer of the GPT-2-124M models incorporating these respective attention mechanisms, trained on the fineweb-10B dataset with a sequence length of 1024 tokens. The final layer is chosen for this analysis as its attention patterns are most indicative of the model's high-level understanding and directly influence the final output. Comparing these maps offers a clear view of how each mechanism synthesises information across the entire sequence.

As shown in Fig. A1, the attention maps for all 12 heads produced by standard Softmax attention (left panel) exhibit the attention sink phenomenon. This is where the first token disproportionately attracts attention from other tokens, irrespective of its semantic importance. This issue is particularly pronounced in longer sequences, where the first token can dominate the attention distribution, leading to suboptimal model performance. In contrast, the attention maps generated by LSSAR (right panel) display a more balanced distribution of attention across tokens. Notably, in the 4x3 grid of LSSAR heads, those in the second column do not exhibit the attention sink phenomenon. This demonstrates that the proposed method effectively mitigates this problem, enabling the model to focus on more relevant tokens throughout the sequence. This visual evidence corroborates our quantitative findings, demonstrating that LSSAR not only enhances length extrapolation capabilities but also improves the overall quality of the attention distributions in transformer models.

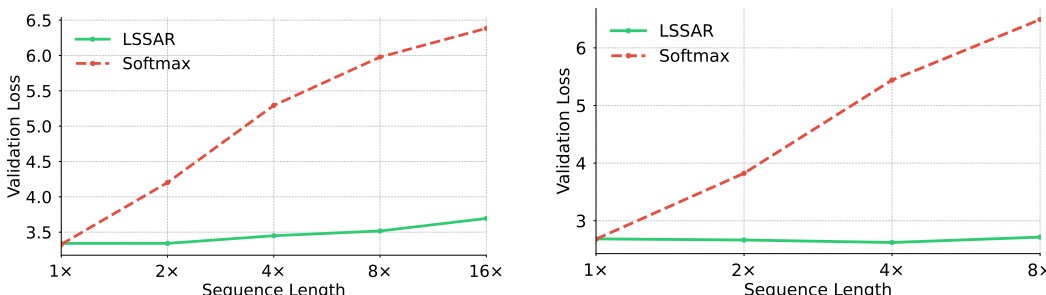

Figure A2: Comparison of Softmax attention and LSSAR($p = 15$) with validation loss extrapolation for GPT-2-45M (left) and GPT-2-355M (right).

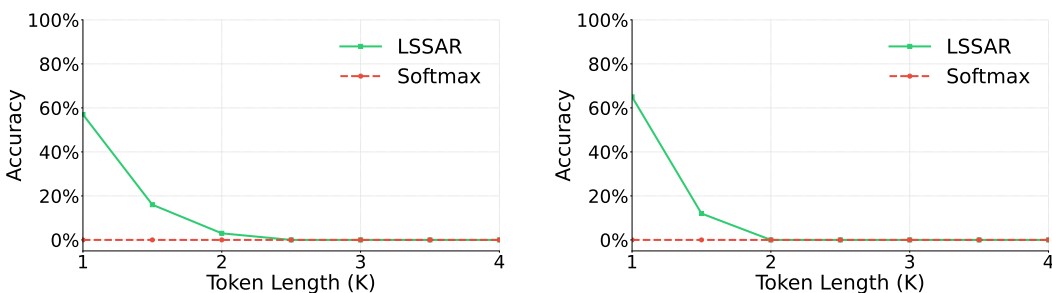

Figure A3: Comparison of Softmax attention and LSSAR($p = 15$) with passkey retrieval accuracy for GPT-2-45M (left) and GPT-2-355M (right). Accuracy was averaged over 100 trials with the passkey placed at random positions within the sequence.

### A.2 EXPERIMENTS FOR SCALING WITH FILTERED DATA

To evaluate the robustness of the proposed LSSAR mechanism across different model scales and data distributions, we conducted a new series of experiments using the FineWeb-Edu dataset (Penedo et al., 2024). We specifically selected FineWeb-Edu over the standard FineWeb dataset used in the main text because it is rigorously filtered using Llama-3-70B, retaining only content with high educational value and logical coherence. This selection serves a dual purpose. On the one hand, it eliminates the potential influence of noisy training data on downstream task performance, ensuring that the evaluation reflects the intrinsic capabilities of the attention mechanism. On the other hand, training on such a clean, high-quality dataset tends to drive models to focus sharply on semantic and syntactic dependencies (Gunasekar et al., 2023). This characteristic paradoxically increases the difficulty of the Passkey Retrieval task, as the model must attend to a "passkey" token that acts as semantic noise within a highly coherent context. In this rigorous setting, the ability of an attention mechanism to distinguish and retrieve the passkey becomes a definitive test of its precision and extrapolation capabilities.

We trained two model configurations from scratch to investigate scaling behaviours. The first is a 6-layer GPT-2-45M model with a configuration suggested by the Pythia suite (Biderman et al., 2023) for analysing scaling behaviours. The second is the standard GPT-2-355M architecture. Both models were modified to incorporate RoPE to ensure a fair comparison with the state-of-the-art extrapolation baseline, and were trained on the FineWeb-Edu dataset with a sequence length of 1024 tokens. The GPT-2-45M model was trained for 9,000 steps, and the GPT-2-355M model was trained for 60,000 steps.

**Validation Loss Extrapolation.** We first evaluated the language modelling performance on sequence lengths extending far beyond the training context. As illustrated in Figure A2, the standard Softmax attention exhibits a significant degradation in validation loss as the sequence length increases, failing to extrapolate effectively even when trained on high-quality data. In contrast, LSSAR maintains a stable and nearly constant validation loss across all tested lengths for both the

45M and 355M models, demonstrating that its entropy invariance property holds true regardless of model scale or data quality.

**Passkey Retrieval Robustness.** The results for the Passkey Retrieval task are presented in Figure A3. Consistent with our hypothesis regarding high-quality training data, the standard Softmax baseline fails completely, yielding 0% accuracy across all tested sequence lengths, including the training window itself. The model's strong bias towards coherent semantic structures prevents it from attending to the random passkey, treating it effectively as noise. In stark contrast, LSSAR successfully overcomes this limitation, achieving substantial retrieval accuracy within the training length (57% for GPT-2-45M and 65% for GPT-2-355M) and maintaining functional capabilities into the extrapolation regime. This result confirms that the proposed re-weighting mechanism effectively sharpens the attention distribution, enabling the model to capture critical high-entropy information even when trained on "textbook-quality" data that discourages such behaviour.

**Downstream Task Performance.** We evaluated zero-shot performance on standard downstream benchmarks. The results are detailed in Table A1. LSSAR consistently outperforms the Softmax baseline across diverse tasks for both model scales. Notably, on the SummScreen benchmark, which requires processing long contexts for summarisation, LSSAR achieves a substantial improvement over Softmax (approximately $4\times$ improvement on GPT-2-355M). These gains confirm that the architectural improvements of LSSAR translate into tangible benefits for complex reasoning and summarisation tasks, without sacrificing generic capabilities.

Table A1: Zero-shot performance on downstream tasks. Best scores are bolded.

| Model | Attention | ARC-E | ARC-C | HellaSwag | PIQA | MMLU | SciQ | SummScreen |
|-------|-----------|-------|-------|-----------|------|------|------|------------|
| GPT-2-45M | Softmax | 45.58 | 16.72 | 27.34 | 58.98 | 22.94 | 70.40 | 0.8100 |
|  | LSSAR($p = 15$) | **46.04** | **18.34** | **27.49** | **59.79** | **22.97** | **75.00** | **2.1932** |
| GPT-2-355M | Softmax | 59.30 | 23.46 | 30.63 | 66.49 | 22.98 | 78.90 | 2.4506 |
|  | LSSAR($p = 15$) | **62.50** | **26.96** | **34.81** | **68.34** | **23.94** | **84.00** | **9.5083** |

**Visualisation of Attention Scores.** Finally, we visualise the attention scores of both models using the same setting as in Sect. A.1. The results are presented in Fig. A4 and Fig. A5. The attention maps produced by standard Softmax attention exhibit the attention sink phenomenon. In contrast, the attention maps generated by LSSAR display a more balanced distribution of attention across tokens. Notably, in the 4x4 grid of LSSAR heads with GPT-2-355M, those in the first and third columns do not exhibit the attention sink phenomenon. This demonstrates that the proposed method effectively mitigates this problem, enabling the model to focus on more relevant tokens throughout the sequence. This visual evidence corroborates our quantitative findings, demonstrating that LSSAR not only enhances length extrapolation capabilities but also improves the overall quality of the attention distributions in transformer models.

### A.3 THE USE OF LARGE LANGUAGE MODELS

In preparing this manuscript, the authors received assistance from large language models (GPT-4o, Gemini 2.5 Pro and Gemini 3 Pro via GitHub Copilot) for the purpose of improving grammar, clarity, and style. The conceptual framework, experimental design, analysis, and conclusions are solely the work of the human authors.

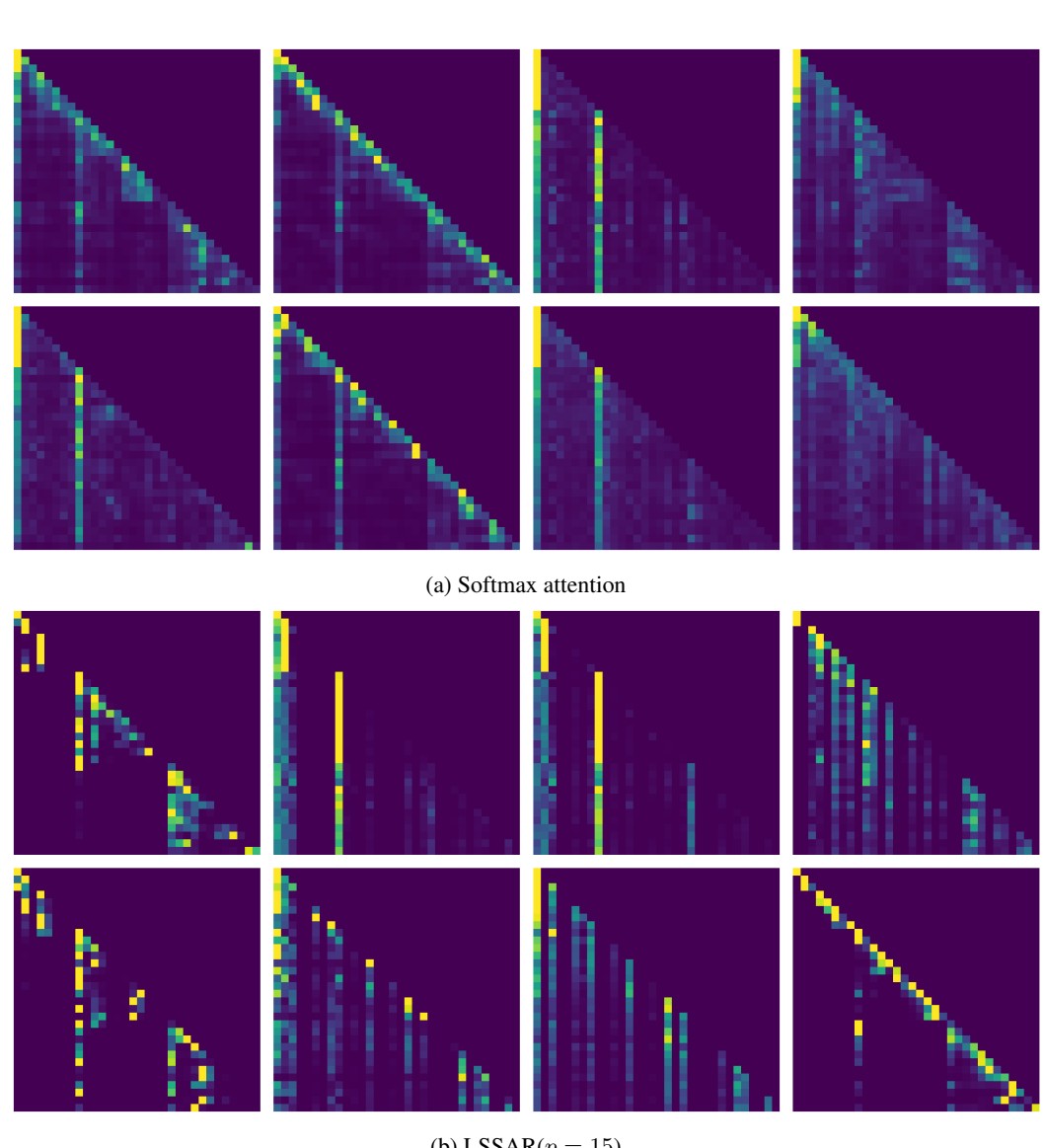

(a) Softmax attention

(b) LSSAR($p = 15$)

Figure A4: Comparison of attention maps from the last layer of GPT-2-45M, showing standard Softmax attention (above) versus LSSAR with $p = 15$ (below). Each panel displays the 8 attention heads in a 4x2 grid. For visualisation purposes, attention scores are clamped to the range $[0, 0.5]$.

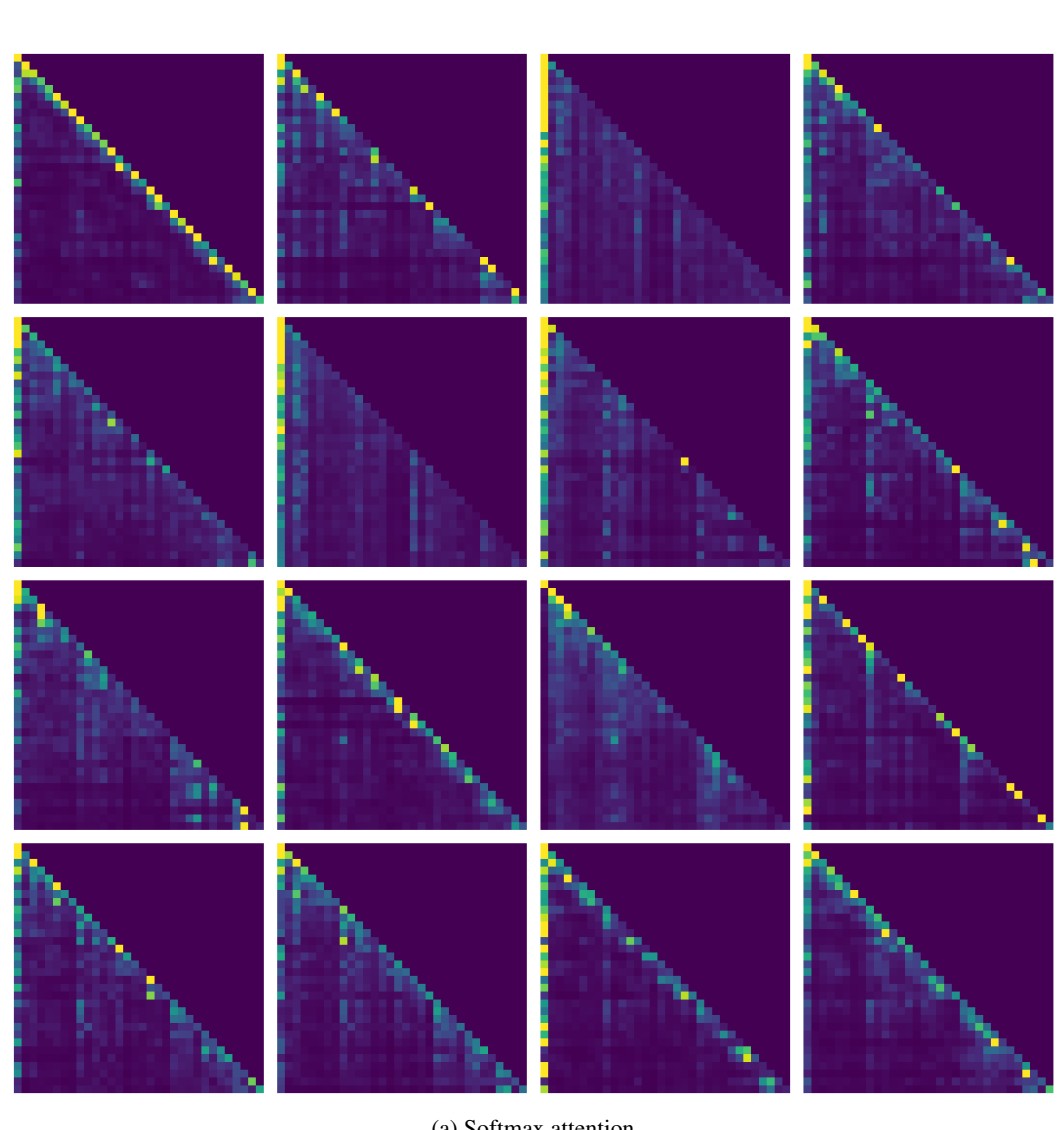

(a) Softmax attention

Figure A5: Comparison of attention maps from the last layer of GPT-2-355M (continued on next page).

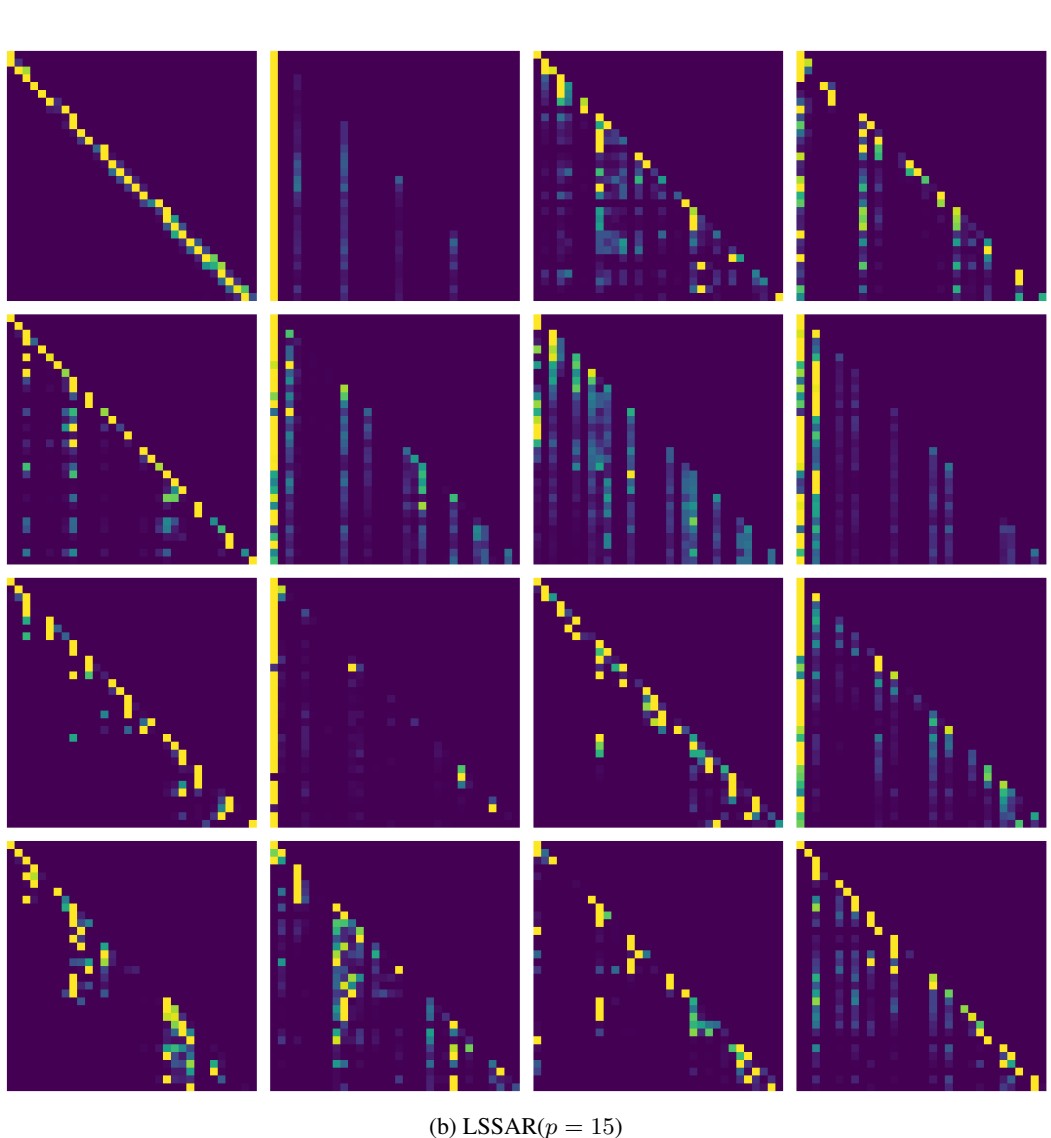

(b) LSSAR($p = 15$)

Figure A5: Comparison of attention maps from the last layer of GPT-2-355M, showing standard Softmax attention (above) versus LSSAR with $p = 15$ (below). Each panel displays the 16 attention heads in a 4x4 grid. For visualisation purposes, attention scores are clamped to the range $[0, 0.5]$.

