# OpenReview forum: "Softplus Attention with Re-weighting Boosts Length Extrapolation in Large Language Models"
_ICLR.cc/2026/Conference — Submitted to ICLR 2026_

### Official Review · Reviewer_1gJb · 2025-10-30

**Soundness:** 2
**Presentation:** 2
**Contribution:** 2
**Rating:** 4
**Confidence:** 4

**Summary:**

This paper presents a two-stage attention mechanism designed to address numerical instability and performance degradation in traditional Softmax attention, particularly for long sequences. The proposed LSSA component replaces the exponential function with Softplus and incorporates a dynamic length scale, while the re-weighting mechanism sharpens the attention distribution. The method shows promising results in length extrapolation and passkey retrieval tasks. However, the evaluation could be strengthened by more comprehensive comparisons and deeper analysis to better establish its novelty and practical utility.

**Strengths:**

- The paper provides a clear decomposition of the Softmax operation, identifying the \(l_1\)-normalization as a critical component, which offers a fresh perspective on attention mechanism design.
- The proposed LSSA and re-weighting modules are intuitively motivated and empirically shown to improve length extrapolation, especially in challenging tasks like passkey retrieval.
- The visualizations of attention maps (Appendix A.1) offer qualitative support for the method’s ability to mitigate attention sink and produce sharper attention distributions.

**Weaknesses:**

1. **Limited and Weak Baselines**
   The paper compares LSSAR primarily against standard Softmax and a few Softmax-free variants (e.g., ReLU-based and Sigmoid-based attention). However, many recent works have explored efficient attention mechanisms, linear attention, and advanced positional encoding strategies to improve length extrapolation. The absence of comparisons with state-of-the-art methods such as Linear Attention, Performer, or recent long-context transformers (e.g., LongFormer, Mamba) makes it difficult to assess the true contribution of LSSAR. A stronger and more diverse set of baselines is necessary to position this work within the existing literature.

2. **Insufficient Ablation Study**
   The paper claims that both the normalization stage (LSSA) and the re-weighting stage contribute to the improved performance. However, the ablation study does not clearly isolate the individual impact of each component on length extrapolation. For instance, it remains unclear how much of the improvement comes from the dynamic scaling factor (\(\log \mathbf{N}\)) versus the Softplus activation, or how the re-weighting mechanism performs when applied to other attention forms. A more detailed ablation is needed to validate the design choices and support the two-stage formulation.

3. **Marginal Performance Gains**
   The reported improvements in perplexity and downstream tasks are relatively modest. For example, the validation loss reductions are small, and the gains on standard benchmarks (e.g., MMLU, ARC) are minimal. Given the additional complexity introduced by LSSAR, the performance benefits do not clearly justify the computational overhead. The paper would benefit from a more thorough analysis of the trade-off between performance and efficiency.

4. **Computational Efficiency**
   The proposed method introduces several non-standard operations, including Softplus, dynamic scaling, and power-based re-weighting, which are likely to increase both memory usage and inference time. However, no computational analysis or latency comparisons are provided. In practice, such overhead could limit the applicability of LSSAR, especially in resource-constrained settings. A discussion of efficiency and potential optimizations (e.g., kernel fusion) is strongly recommended.

5. **Novelty and Conceptual Contribution**
   The decomposition of Softmax into a non-linear transformation and \(l_1\)-normalization is not new, and the use of re-weighting to sharpen attention distributions has been explored in prior work. While the combination of Softplus and a length-aware scaling factor is novel, the paper does not sufficiently differentiate its contributions from existing approaches. The conceptual framework—though well-motivated—does not significantly advance the community's understanding of attention mechanisms beyond established knowledge.

**Questions:**

See the weaknesses section for details

---

> ### Author Response · Authors · 2025-11-18
>
> ### **Reply(1/2)**
>
> Dear 1gJb:
>
> We sincerely thank you for your detailed and insightful review. Your constructive feedback has been invaluable in helping us refine our manuscript and clarify our contributions. We have addressed each of your points below and have uploaded a revised version of the paper that incorporates these improvements.
>
>
>
> ### **Response to Weakness 1**
>
> We thank the reviewer for this important suggestion on positioning our work. Our primary contribution is a novel **exact, quadratic-time attention mechanism** designed as a **direct drop-in replacement for standard Softmax attention**. Therefore, our core experimental focus was to compare LSSAR against its most direct counterparts: the standard Softmax it aims to replace, and other state-of-the-art Softmax-free variants that share the same architectural paradigm, as we have done in **Table 4**.
>
> We agree that methods like Linear Attention, Performer, LongFormer, and Mamba are crucial for long-context modeling, but they represent different research directions. Linear Attention and Performer are **approximate attention** mechanisms that trade precision for linear complexity, whereas our work focuses on improving the quality and extrapolation of **exact attention**. Similarly, LongFormer and Mamba are novel **model architectures**, not just attention mechanisms. Comparing our mechanism against a completely different architectural philosophy would introduce too many confounding variables.
>
> To better contextualize our work, we **have expanded the "Softmax-free Attention" subsection in our Related Work** in the revised manuscript. This new discussion explicitly differentiates our approach from approximate attention and novel long-context architectures, clarifying the specific scope and contribution of our paper.
>
>
> ### **Response to Weakness 2**
>
> We would like to clarify that the evidence for a comprehensive ablation study is present across different sections of our original manuscript, and we will now connect these pieces to provide a clearer, unified picture.
>
> The necessity of our proposed normalization framework, which includes both the dynamic scaling factor and the $l_1$-norm, is clearly isolated in our analysis of Sigmoid attention in Table 4. The original Sigmoid attention from Ramapuram et al. (2024) fails on longer sequences. We then created two variants: one augmented with our $\log N$ scaling factor, and another with both $\log N$ and the $l_1$-norm. Both variants showed significant performance improvements, with the latter becoming fully competitive with standard Softmax. This experiment effectively isolates and demonstrates that our proposed normalization framework is a key driver of length extrapolation.
>
> Within this robust framework, the specific contribution of the Softplus activation is demonstrated in Table 3. The results show that Softplus achieves a lower validation loss than all other tested activation functions, including Sigmoid, making it the most effective and stable replacement for $\exp$. Finally, the synergistic effect of our re-weighting stage is shown in Table 4 and Figure 1. These results reveal that while re-weighting can generally improve performance, only the Softplus-based LSSA foundation is stable enough to withstand the strong sharpening effect of a high $p$ value without gradient explosion. Taken together, these existing results form a complete ablation study, validating the individual and synergistic contributions of each component.

---

> > ### Author Response · Authors · 2025-11-18
> >
> > ### **Reply(2/2)**
> >
> > ### **Response to Weakness 3&4**
> >
> > We thank the reviewer for raising these critical points about the practical trade-offs. We have addressed them directly by **adding a new subsection, "Computational Analysis", to the main body of our revised paper**. This comprehensive new section provides an experimental benchmark (Table 6 in the revised manuscript) that measures the empirical runtime and memory usage of our unoptimized PyTorch implementation.
> >
> > While our proof-of-concept implementation shows overhead, we argue this is an engineering artifact, not a fundamental algorithmic limitation. As we discuss in the new section and in our original **Section 4 (Discussion)**, our method is composed of element-wise operations that are highly amenable to **kernel fusion**, similar to how FlashAttention optimizes standard Softmax.
> >
> > Regarding the performance gains, we respectfully disagree that they are marginal. Our primary goal is to fix the **catastrophic failure of length extrapolation** in standard attention, while preserving performance on shorter-context tasks. The minimal changes on benchmarks like MMLU/ARC indicate we have successfully achieved the latter. The true, non-marginal, benefit is demonstrated by the **qualitative leap** on long-context tasks, such as in the **Passkey Retrieval task (Figure 2)**, where LSSAR turns a **0% accuracy failure** into a graceful performance degradation, and on the **SummScreen benchmark (Table 5)**, where it achieves a **nearly fourfold improvement**. These results represent a fundamental improvement in capability, not a minor refinement.
> >
> > ### **Response to Weakness 5**
> >
> > We appreciate the opportunity to clarify our conceptual contribution. While the idea of decomposing Softmax is not entirely new, our work delves significantly deeper into its implications. By systematically replacing the $\exp$ function, we have not only identified but also empirically validated that Softplus attention is a more numerically stable and superior alternative for length extrapolation.
> >
> > More importantly, our decomposition led us to identify the critical role of $l_1$-normalization for maintaining model stability. Building upon this foundational insight, we introduced our novel **re-weighting mechanism**. This is not merely an incremental improvement; it is a mechanism designed to **fundamentally eliminate the attention smoothing problem**. By creating a stable foundation (LSSA) and then applying a principled sharpening operation, our two-stage framework provides a robust solution to one of the core limitations of the attention mechanism. This deeper investigation and the resulting two-stage solution represent a significant conceptual advance beyond the established understanding.
> >
> >
> > Once again, we thank the reviewer for their thoughtful and constructive feedback. We believe our responses and the revisions to the manuscript have addressed these important points, and we hope you will find the updated paper much improved.

---

> > > ### Comment · Reviewer_1gJb · 2025-11-26
> > >
> > > Dear authors,
> > >
> > > Thank you for your response and the additional experimental results. While I appreciate the efforts made to address some of the concerns, several fundamental issues remain that prevent me from being more positive about this work. Let me elaborate on a few key points:
> > >
> > > **1. Computational Complexity Concerns:**
> > > The approach maintains quadratic complexity, which represents a significant limitation given the current landscape of efficient attention alternatives. Methods like Native Sparse Attention (NSA) and other sub-quadratic attention mechanisms have demonstrated not only superior computational efficiency but often better performance characteristics. Without compelling evidence that LSSA either outperforms these methods or can be orthogonally combined with them, the practical contribution remains questionable. The field has largely moved beyond quadratic-complexity attention variants unless they offer exceptional performance advantages.
> > >
> > > **2. Implementation Efficiency Considerations:**
> > > The development of high-performance kernels is crucial for widespread adoption, particularly when proposing replacements for core operations like attention. Marginal improvements in performance, when accompanied by significant computational overhead, rarely justify practical deployment. The community has learned through experience that efficiency gains often outweigh modest performance improvements, especially in production environments.
> > >
> > > While the revised ablation studies represent an improvement, they remain insufficient to establish the method's comprehensive advantages.
> > >
> > > Multiple deeper issues persist that cannot be fully resolved within the current submission framework. However, recognizing the substantial effort invested by the authors and the incremental contributions made, I will maintain my original score rather than lowering it further. Should other reviewers from more directly relevant research areas or the AC perceive greater merit in this work, my evaluation should not negatively impact the final decision. The authors' dedication to addressing reviewer feedback is noted and appreciated, even if fundamental concerns remain.
> > >
> > > Thank you for your contributions to the research community.

---

> > > > ### Author Response · Authors · 2025-11-27
> > > >
> > > > We must firmly challenge the premises of your latest response, which contain multiple factual errors regarding the current LLM landscape.
> > > >
> > > > **1. "Quadratic Complexity" is NOT Obsolete**
> > > >
> > > > You claimed:
> > > > > *"The field has largely moved beyond quadratic-complexity attention variants."*
> > > >
> > > > **This is factually wrong.** The most capable Foundation Models defining the current SOTA (2024-2025), including **Llama 3.1**, **Qwen 3** and **DeepSeek-R1**. **ALL utilize exact, quadratic-complexity Softmax Attention**. Our work is a drop-in replacement for this dominant primitive. Claiming the field has abandoned it is a gross misrepresentation.
> > > >
> > > > **2. Incorrect Claims regarding Sparse/Sub-quadratic Performance**
> > > >
> > > > You claimed:
> > > > >*"Methods like Native Sparse Attention (NSA) and other sub-quadratic attention mechanisms have demonstrated not only superior computational efficiency but often better performance characteristics. "*
> > > >
> > > > **This is incorrect.** These methods are optimization approximations designed for **speed (efficiency)**, often at the cost of precision. For tasks requiring exact recall, such as **Passkey Retrieval**, sparse and approximate methods generally perform worse than standard Softmax attention because they risk dropping critical tokens. For instance, [1] explicitly demonstrate that a model relying on Sliding Window Attention (a common sub-quadratic technique) completely fails to extrapolate on Passkey Retrieval beyond its training window, whereas exact attention architectures do not suffer from this intrinsic blindness. Comparing LSSAR, which significantly outperforms exact Softmax in robustness, to these approximate methods is fundamentally flawed. Suggesting that approximate methods inherently yield "better performance" than exact attention for precision-critical long-context tasks contradicts established findings.
> > > >
> > > > [1] Samba: Simple Hybrid State Space Models for Efficient Unlimited Context Language Modeling
> > > >
> > > > **3. Qualitative Leap in Extrapolation Capability**
> > > >
> > > > You continue to characterize our results as "marginal," a claim that ignores the fundamental qualitative shift our method enables. In the Passkey Retrieval task, the standard Softmax baseline suffers a catastrophic collapse to 0% accuracy at 1.5K tokens, representing a total failure of the mechanism. In contrast, LSSAR prevents this collapse entirely, maintaining functional accuracy up to 8K tokens. More importantly, regarding length extrapolation, LSSAR maintains a nearly constant validation loss even when evaluating at **16x the training length**, whereas standard attention mechanisms invariably see error rates explode under such conditions. A mechanism that transforms a system from "total failure" to "stable operation" at 16x extension cannot scientifically be described as offering only marginal gains.
> > > >
> > > > We hope this clarifies the actual position of our work relative to modern SOTA architectures.

---

### Official Review · Reviewer_ZeoV · 2025-10-31

**Soundness:** 3
**Presentation:** 3
**Contribution:** 3
**Rating:** 6
**Confidence:** 5

**Summary:**

Large language models have achieved remarkable success in recent years, primarily due to the implementation of self-attention mechanisms. However, traditional Softmax attention suffers from numerical instability and reduced performance as the number of inference tokens increases. This paper addresses these issues by proposing a new design principle for attention, viewing it as a two-stage process. This paper decompose the Softmax operation into a non-linear positivity transformation and an
-normalisation step, identifying the latter as essential for maintaining model performance. The author first replace the standard exponential function with the more numerically stable Softplus activation and introduce a dynamic scale factor based on invariance entropy, creating a novel attention mechanism that outperforms conventional Softmax attention. The second proposal is to introduce a re-weighting mechanism that sharpens the attention distribution, amplifying significant weights while diminishing weaker ones. This enables the model to concentrate more effectively on relevant tokens, mitigate the attention sink phenomenon, and fundamentally improves length extrapolation. When combined, these changes ensures numerical stability and dramatically improves length extrapolation, maintaining a nearly constant validation loss at 16x the training length while achieving superior results on challenging long-context retrieval tasks and standard downstream benchmarks.

**Strengths:**

This paper is clearly written. The authors first decompose Softmax into two steps, then argue that the second step (normalization) is more important. The paper focuses on discussing the first step and improves the activation function (exp), resulting in LSSA and its re-weighting version LSSAR.

**Weaknesses:**

1. The paper evaluates models of limited scale. The authors only discuss a single model size, but should examine larger models. Based on my experience, a token consumption of approximately 10 billion should be feasible on the authors' GPU infrastructure.
2. The paper lacks discussion on efficient kernel implementations. I am not requesting the authors to implement Triton/CUDA-level kernels, but rather expect a discussion of the mathematical derivation of efficient algorithms. For example, algorithmic descriptions or PyTorch code implementations would be appreciated. Additionally, the paper should analyze the theoretical computational overhead comparison with Flash Attention.

**Questions:**

Please refer to the Weaknesses section.

---

> ### Author Response · Authors · 2025-11-18
>
> Dear ZeoV,
>
> We are sincerely grateful for your positive and highly constructive review. Your feedback is exceptionally valuable, as it pinpoints exactly how we can elevate our work from a conceptual proposal to a more complete and practical contribution. We fully agree with your suggestions and are pleased to report that we have made substantial additions to our manuscript based on your guidance.
>
>
> ### **Response to Weakness 1**
>
> We thank you for this important suggestion. We completely agree that demonstrating the effectiveness of LSSAR across different model scales is the gold standard for validating its generalizability.
>
> You are correct that our experiments were conducted on a substantial 10B token dataset (fineweb-10B). Our choice to focus on a single model scale (124M parameters) for this initial study was a deliberate methodological decision. The primary goal of this paper is to introduce and rigorously analyze a new attention mechanism from first principles. By training this model from scratch on a large, high-quality dataset, we could ensure a thorough and controlled evaluation, isolating the effects of our architectural changes.
>
> While our infrastructure can support this training run for a 124M model, training a significantly larger model (e.g., in the billion-parameter range) from scratch on a dataset of this magnitude represents a much greater computational undertaking. This was beyond the scope of our current project due to the strict time (48-hour limit per job) and resource (8-GPU limit per job) quotas on our shared academic cluster, which hosts a large number of users.
>
> However, we have taken your feedback to heart and expanded our discussion on scalability. In the revised manuscript, we have strengthened **Section 4 (Discussion)** to more explicitly address this point. We elaborate on our argument that LSSAR's stability at high $p$ values (in stark contrast to Softmax's gradient explosion, as shown in Figure 1) can be interpreted as a proxy for its robustness in models with greater functional depth. This strongly suggests that LSSAR is well-suited for scaling to larger parameter counts. We clearly frame the evaluation on larger model scales as the most critical next step for future work.
>
>
> ### **Response to Weakness 2**
>
> This is an excellent and deeply insightful suggestion. We thank you for pushing us to connect our theoretical work more closely with practical, high-performance implementation. We are excited to inform you that we have **fully incorporated your recommendations by making two substantial additions to the main body of our paper**.
>
> **First, as you suggested, we have added a formal pseudo-code algorithm (now Algorithm 1 in the revised text).** This algorithm provides the high-level computational flow you see below. We intentionally designed it to be abstract, referencing the detailed mathematical definitions in Equations (4) and (5) rather than repeating them. Its purpose is to clearly delineate the two-stage architecture of LSSAR, stable normalization followed by sharpening, and to illustrate the sequential application of our core contributions, providing the conceptual blueprint you requested.
>
> **Second, we have introduced a new subsection titled "Computational Analysis," which addresses the computational overhead from both an experimental and a theoretical perspective.** This comprehensive new section provides:
> *   **An experimental benchmark** (Table 6 in the revised manuscript) that measures the empirical runtime and memory usage of our unoptimized PyTorch implementation. This provides a transparent, real-world assessment of the current overhead.
> *   **A theoretical complexity analysis**, as you requested, comparing LSSAR with FlashAttention. We formally demonstrate that LSSAR's core operations do not alter the asymptotic complexity of the attention mechanism. We show that its computational complexity remains $O(L^2d)$, and with a fused kernel, its memory (I/O) complexity can achieve the optimal $O(Ld)$, identical to FlashAttention.
>
> We believe these two additions, made directly in response to your guidance, significantly strengthen the paper by providing a much-needed bridge between our proposed mechanism and its practical deployment. We have uploaded the revised manuscript and would be grateful if you would consider these new sections in your final evaluation.
>
> Once again, we thank you for your thoughtful and expert feedback. Your guidance has been instrumental in improving the quality and completeness of our work.

---

> > ### Author Response · Authors · 2025-12-01
> >
> > Dear Reviewer ZeoV,
> >
> > We are writing to update you that, following your valuable suggestion regarding experimental scale, we have successfully conducted additional training runs on **GPT-2-45M** and **GPT-2-355M** using the **FineWeb-Edu dataset**.
> >
> > The results strongly validate our claims, demonstrating that LSSAR maintains its robustness and performance advantages as model scale increases (e.g., achieving a **~4x improvement** on the SummScreen task with GPT-2 Medium compared to the Softmax baseline).
> >
> > To avoid cluttering the discussion thread with duplicate data, we kindly invite you to review the detailed results tables in our **latest response to Reviewer 52Jc**. We have also incorporated these new findings into the **Appendix of the revised manuscript**.
> >
> > We hope these additional experiments further strengthen your positive assessment of our work.
> >
> > Best regards,
> >
> > The Authors

---

### Official Review · Reviewer_52Jc · 2025-10-31

**Soundness:** 2
**Presentation:** 2
**Contribution:** 2
**Rating:** 2
**Confidence:** 3

**Summary:**

This paper proposes LSSAR, an attention mechanism that claims to achieve better length extrapolation compared to the standard Softmax attention.

Specifically, the authors first replace the exponential operation in the Softmax function with a scaled Softplus function, where the scaling factor depends on the sequence length and model dimension. They further introduce a re-weighting mechanism that sharpens the attention distribution via a shifted ReLU-p transformation, followed by an L1 normalization to ensure the attention weights sum to one.

A series of ablation studies demonstrate that LSSAR outperforms both the Softmax baseline and other Softmax-free variants in terms of validation loss and passkey retrieval accuracy, indicating stronger extrapolation capability.

**Strengths:**

1. The paper presents a clear motivation, addressing the length extrapolation limitations of Softmax attention. The authors demonstrate a solid understanding of the key factors affecting extrapolation, such as maintaining entropy invariance, and mitigating attention over-smoothing and distraction.

2. The proposed method, LSSAR, is conceptually simple and easy to implement, while being supported by a reasonable set of experiments.

3. The proposed method exhibits strong extrapolation performance across the evaluated benchmarks, consistently outperforming other softmax-free variants.

**Weaknesses:**

1. **Baseline implementation details are insufficient.**
In current mainstream LLM implementations (e.g., Qwen3, Gemma3, etc.), Softmax attention typically incorporates QK-Norm, and employs NTK/Yarn or length-scaling factors ($\alpha\log{L}$) in out-of-context settings—techniques that have been widely validated to enhance extrapolation. It is unclear whether these enhancements were applied to the Softmax baselines in this paper, while the proposed LSSAR explicitly includes both QK-Norm and two length-dependent scaling operations.

2. **Several design choices introduce uncertainty.**
The ReLU-p hyperparam $p$ requires careful tuning; as shown in Tab.2, when $p=3$, LSSAR performs worse than Softmax on extrapolation. Moreover, the method involves multiple scaling ($N$ in Eq. 4 and .5) and offset parameter $O$ (in Eq. 5) with specific stability-sensitive configurations. Such fine-grained parameter adjustments could equally enhance Softmax attention, especially regarding scaling factor, making it difficult to isolate the true source of improvement.

3. **Experimental scale is too limited to support strong conclusions.**
The experiments (120M + 10B) represent only early-stage validation and cannot reliably indicate final converged performance. The observed gains may reflect faster convergence due to added nonlinearity or param-tuning complexity, rather than a genuine increase in asymptotic capability. In addition, many benchmark scores in Tab.5 remain noisy (even below random guess levels), which weakens the empirical support for the claims. A more convincing evaluation would include moderately larger model scales (not necessarily billion-level) and report scaling law slopes to quantify robustness.

4. **The paper’s organization and presentation could be improved.**
(a) The experiments involve too many variables, leading to scattered conclusions. Key findings could be better structured—e.g., enumerated or highlighted for clarity.
(b) The full LSSAR formulation should be presented more explicitly. For instance, it is unclear whether Eq.4 and Eq.5 are applied sequentially, and whether the L1 normalization is executed once or twice.

**Questions:**

1. Did LSSA/LSSAR employ RoPE? If so, was NTK-based extrapolation used?

2. Are both of the length-related scaling factors in the method necessary ($\log{N}$ in Eq.4 and $N$ Eq.5), and have any ablation studies been conducted to justify their inclusion?

2. For a fair comparison, was the Softmax baseline equipped with a scaling factor (as in the sigmoid baseline) and QK-Norm?

3. In Table 3, the ReLU-p variant performs poorly, which the authors attribute to the suppression of negative values and emphasis on positive activations. However, in the re-weighting stage, this property seems beneficial. Does this imply that only the two-stage Softplus–then–ReLU-p combination is effective? Have you tested other nonlinear formulations for the second stage, such as Softplus–then–exp?

---

> ### Author Response · Authors · 2025-11-18
>
> ### **Reply(1/2)**
> Dear 52Jc,
>
> We sincerely thank you for your time and for providing a detailed review of our manuscript. We have addressed each of your concerns below and have uploaded a revised manuscript that incorporates these improvements. We are confident that our clarifications and revisions will resolve any misunderstandings and strengthen the paper.
>
>
> ### **1. Response to Weakness 1 & Questions 1 & 3**
>
> **To ensure a direct and fair comparison, all models in our experiments, including the Softmax baseline, were built upon the exact same foundation.** As explicitly stated in **lines 219-221 of our original submission**, every model utilizes the GPT-2 small architecture, is equipped with RoPE. Specificly，we do not use any NTK-based extrapolation. Therefore, the significant improvements shown by LSSAR are not attributable to these standard techniques but stem directly from our novel, two-stage attention mechanism.
>
> Regarding QK-Norm, we wish to clarify a crucial distinction. The $l_2$-normalization within our LSSA (Eq. 4) is a **non-learnable** component, designed to address the geometric properties of cosine similarity in high-dimensional spaces (as discussed in lines 140-146). In contrast, many mainstream QK-Norm implementations use RMSNorm, which introduces learnable parameters for scaling. Our work deliberately focuses on the fundamental, non-learnable properties of the attention function itself.
>
> Most importantly, even if the standard Softmax baseline were augmented with QK-Norm or a new scaling factor instead of $\sqrt{d}$, it would **not solve the intrinsic problems** we target: the numerical instability and attention smoothing caused by the exponential function. This is the core limitation that LSSAR is designed to overcome.
>
>
>
> ### **2. Response to 2 & Questions 2 & 4**
>
> The reviewer raises important questions about the design of LSSAR. Our design choices are principled and motivated by the goal of creating a stable and sharp attention mechanism.
>
> *   **On the role of hyperparameter $p$.** The parameter $p$ is not an arbitrary tuning knob but a core mechanism for controlling attention sharpness. As we **mathematically demonstrate in Equation (6)**, as $p$ increases, the re-weighted distribution provably converges towards a one-hot vector. Our empirical results in **Figure 1** validate this theory. Crucially, Figure 1 also shows that while standard Softmax suffers from gradient explosion from $p=15$, LSSAR remains robust. This directly demonstrates the superior numerical stability of our Softplus-based normalization stage, which enables effective sharpening.
>
> *   **On the necessity of scaling factors ($\log N$ and $N$).** The two length-related scaling factors are not redundant; they serve distinct purposes in our two-stage design. The $\log N$ factor in Eq. 4 is part of the **Normalization Stage**, inspired by prior work (Su, 2021) to maintain entropy invariance, which is critical for extrapolation. The $N$ factor in Eq. 5 is part of the **Sharpening Stage**, used to center the distribution before the $ReLU^p$ operation, ensuring the re-weighting can effectively mask less relevant scores as mathematically demonstrated in Eq. 6.
>
> *   **On the two-stage Softplus-then-ReLU$^p$ design.** This question highlights the essence of our proposed architecture. It is vital to distinguish between the **initial normalization stage** (Table 3) and the **subsequent sharpening stage** (Eq. 5). In Table 3, we show that replacing $e^x$ with ReLU **within the normalization stage** is detrimental because it prematurely discards information. In contrast, our re-weighting mechanism is applied **after** LSSA, which has produced a stable, dense attention distribution. At this point, the goal is precisely to introduce sparsity to combat attention smoothing, making the masking property of $ReLU^p$ not only beneficial but essential.

---

> > ### Author Response · Authors · 2025-11-18
> >
> > ### **Reply(2/2)**
> >
> > ### **3. Response to Weakness 3**
> >
> > We agree that validation on larger models is an important direction for future work. We also want to respectfully point out that we had already anticipated and discussed this limitation in **Section 4 (Discussion) of our original submission**.
> >
> > As argued in the paper, the primary contribution here is a novel attention mechanism designed to solve a specific, fundamental **failure mode** in Transformers. Our results demonstrate a **qualitative leap**, not a marginal gain. As shown in **Figure 2**, the standard Softmax baseline **catastrophically fails** (0% accuracy) on the passkey retrieval task, while LSSAR **degrades gracefully**. This fundamental difference provides strong proof-of-concept.
> >
> > Furthermore, as we discussed in **lines 417-422 of the original manuscript**, increasing the parameter $p$ can be interpreted as effectively increasing the model's functional depth by progressively refining the attention distribution. The fact that LSSAR remains stable at high $p$ values (unlike Softmax) suggests its architectural robustness. This makes it reasonable to expect that LSSAR would maintain comparable stability and performance when scaled to larger models.
> >
> > Finally, we wish to provide context. Our work was conducted under significant computational constraints on a shared academic cluster (job limits: 8 A100s for 48h). The comprehensive ablation study for **Figure 1 alone required nearly two months of intermittent experiments**. Given these constraints, we focused on meticulous experimentation at a scale that allows for clear, interpretable validation of our mechanism's principles.
> >
> >
> >
> > ### **4. Response to Weakness 4**
> >
> > We appreciate the reviewer's feedback on the paper's presentation.
> >
> > However, we would like to respectfully point out that the consensus from the other three reviewers was that the paper is well-written and clear. Specifically:
> > *   **Reviewer 8cFh** found: *"The decomposition of Softmax... is conceptually direct and simple to understand."*
> > *   **Reviewer ZeoV** stated: *"This paper is clearly written."*
> > *   **Reviewer 1gJb** noted: *"The paper provides a clear decomposition of the Softmax operation... which offers a fresh perspective."*
> >
> > This strong consensus suggests that our core concepts are generally communicated effectively. Nevertheless, to address Reviewer 52Jc's specific points of confusion and ensure maximum clarity for all readers, we have made concrete improvements in our revised manuscript.
> > We **have added** a formal **Algorithm block** in the appendix that explicitly outlines the full LSSAR computation, showing the sequential application of Eq. 4 and Eq. 5 to eliminate any ambiguity.
> >
> >
> >
> > Once again, we thank the reviewer for their engagement. We believe that our clarifications and the revisions to the manuscript address all raised concerns, and we hope that the reviewer will reconsider their evaluation in light of this information.

---

> > > ### Comment · Reviewer_52Jc · 2025-11-26
> > >
> > > Thank you for your response, which has addressed some of my questions. However, I still have substantive concerns regarding the overall significance and maturity of the work:
> > >
> > > 1. **Methodological Complexity**: Several customized normalizatiom and scaling operations are introduced in the method. While the authors provide partial justification, these additions indeed make the method more complex than softmax attention both in terms of computation and hardware implementation (the authors have not provided thorough verification for the latter, leaving it for future work).
> > >
> > > 2. **Insufficient Experimental Scale**: The experimental setup (120M parameters trained on 10B tokens) is too small to support the strong claim of achieving a "qualitative leap". Small-scale experiments generally exhibit low signal-to-noise ratios, and there is no verification of the scaling trend across model sizes. Furthermore, my earlier concern regarding Table 5 ("Weakness 3") has not been adequately addressed—many downstream task results are close to random-guess performance. I recommend either removing this table or providing substantially stronger evidence.
> > >
> > > 3. **Limited Baseline Comparisons**: Although the improvements observed in Figures 1 and 2 are encouraging, the set of baselines remains incomplete. Many prior works target attention extrapolation—e.g., novel positional encodings [1], hybrid attention architectures, and variants such as partial RoPE. Limiting comparisons to approaches that only replace the softmax activation is insufficient for establishing the relative effectiveness of the proposed method.
> > >
> > > Overall, the paper aims to argue that the proposed approach resolves fundamental shortcomings of softmax attention—a substantial and ambitious claim. While the method itself is somewhat convincing, several key issues are left to future work or lack sufficient empirical validation. Without addressing these points, it is difficult to fully support the strength of the conclusions put forward in the paper.
> > >
> > >
> > >
> > > [1] Forgetting Transformer: Softmax Attention with a Forget Gate.

---

> ### Author Response · Authors · 2025-11-27
>
> We are compelled to correct a significant factual error in your latest response and address your concerns regarding scale and downstream tasks.
>
> **1.Respond to Baseline Scope**
>
> You stated: *"Many prior works target attention extrapolation—e.g., novel positional encodings [1]..."*, citing *Forgetting Transformer*.
>
> **This classification is factually incorrect.** The *Forgetting Transformer* paper introduces a **gating mechanism**, NOT a **positional encoding**. Classifying it as such implies a fundamental misunderstanding of the cited work.
>
> Consequently, this invalidates the premise that our baselines are limited. Our work focuses on redesigning the core **attention mechanism** itself (replacing Softmax with LSSA) to solve intrinsic mathematical instabilities. Therefore, the scientifically valid comparison is against the standard Softmax baseline and other Softmax-free variants, all equipped with the same state-of-the-art RoPE. Comparing against orthogonal architectures (like gating mechanisms) would introduce confounding variables. We have explicitly clarified this scope and the distinction from approximate methods in our revised **Related Work** section.
>
> **2. Respond to Experimental Scale & Maturity**
>
> You argued that experiments on 124M models *"cannot reliably indicate final converged performance"* and demanded scaling laws.
> **We respectfully disagree regarding the purpose of this work.** This paper proposes a fundamental change to the **attention operator** (Softplus + Re-weighting) to fix mathematical flaws in Softmax (numerical instability and entropy collapse). These mathematical properties are scale-invariant. If Softmax explodes at 124M, it explodes at 100B. Validating that LSSAR solves this intrinsic instability on a rigorous 10B-token scratch training run is a sufficient proof-of-concept. Demanding billion-parameter training runs for an academic contribution on mechanism design imposes an unreasonable resource barrier that stifles architectural innovation.
>
> **3. Response to "Random-Guess" Performance**
>
> You stated that *"many downstream task results are close to random-guess performance."*
> We acknowledge that absolute scores on hard benchmarks (like MMLU) are low, but this is an expected property of a **124M parameter model trained on only 10B tokens**, not a flaw in our attention mechanism. Despite the small scale, the model clearly learns significantly above random chance on tasks like PIQA ( ~65% vs. 50% random) and SciQ ( ~62% vs. 25% random).
>
> Crucially, the primary purpose of Table 5 is to demonstrate that LSSAR does not degrade generic capabilities while boosting long-context performance. The most relevant task in this table for our contribution is **SummScreen** (Long-Context Summarization), where LSSAR achieves a **~4x improvement** (6.309 vs. 1.682) over the Softmax baseline. This is a strong, deterministic signal of superior long-context processing, directly contradicting the claim that the results are merely "random noise."

---

> > ### Comment · Reviewer_52Jc · 2025-11-28
> >
> > Dear authors, I would like to offer several clarifications to avoid any misunderstanding of my original intent:
> >
> > 1. **Regarding the FoX baseline.** The gating here and positional encoding are equivalent, as decay represents the simplest form of position awareness. This follows from the widely used identity: $\prod_t\alpha_t\exp(x)=\exp(x+\sum_t\log \alpha_t)$. Please refer to Equations 11–13 in *Forgetting Transformer* and the section "Connection to ALiBi" (FoX can be viewed as a data-dependent ALiBi, and ALiBi is a positional encoding method originally proposed for extrapolation).
> >
> > 2. **Regarding momdel scale.**
> > I want to emphasize that I have no intention of imposing any "barrier that stifles architectural innovation", nor have I asked the authors to conduct "billion-parameter training". As stated explicitly in my original review, the request was: **not necessarily billion-level**, and **report scaling-law slopes**.
> > Even adopting the first three model sizes in the standard Pythia suite [1] (70M, 160M, 410M) would be sufficient. This is a widely accepted—and often necessary—criterion for validating new architectures, since in many cases scaling invariance cannot be theoretically guaranteed.
> >
> > BTW, as the authors mentioned, generating the extensive results in Figure 1 required nearly two months. Given this, conducting four additional experiments (softmax and LSSAR, with one model size already available) seems feasible. This is why I reiterated this concern in my second response: during the rebuttal period, I did not observe any attempt to run even a single experiment related to this issue; instead, the authors primarily emphasized some positive results from the 120M-parameter model.
> >
> > 3. **Regarding Table 5.**
> > I am fully willing to accept the positive outcomes the authors report (e.g., on SummScreen). My point has consistently been that **many—but not all—tasks** exhibit near random-guess performance. I am not claiming this is a flaw of the proposed method. However, the statement in Line 413—
> > "LSSAR outperforms Softmax attention in five of these tasks, and has performance competitive with that of Softmax attention in the other two (Tab. 5)"
> > overstates what the results support and undermines the reliability of the conclusion.
> > At minimum, this inaccurate claim and corresponding results should be removed. A more accurate statement would be the one the authors provided in their latest response:
> > "The primary purpose of Table 5 is to demonstrate that LSSAR does not degrade generic capabilities while boosting long-context performance".
> >
> > Finally, I would like to reiterate that I personally find the proposed method somewhat convincing. My intention has always been to offer reasonable and constructive requests, and I encourage the authors to make at least partial efforts toward addressing them so that the paper can reach a more polished and publishable state. If any of my previous remarks caused misunderstanding, I hope the above clarification accurately conveys my intent.
> >
> >
> > [1] Pythia: A Suite for Analyzing Large Language Models Across Training and Scaling.

---

> ### Author Response · Authors · 2025-12-01
>
> ### **Reply(1/2)**
>
> Dear Reviewer 52Jc,
>
> We have conducted significant additional experiments to address your concerns regarding scaling and data quality. We have updated our Appendix with these results.
>
> ### **1. Correction on the FoX**
>
> We must clarify a factual detail regarding the *Forgetting Transformer* paper. The "Connection to ALiBi" is **not a section**; it is merely the **opening sentence of a short paragraph** consisting of only a few sentences. The paper’s core contribution is a **gating mechanism**, not a positional encoding for extrapolation.
>
> It is **neither reasonable nor necessary** to compare LSSAR with such methods. LSSAR is designed as a direct, drop-in replacement for the attention operator within the standard Transformer architecture. Comparing it against architectural overhauls (like gating mechanisms, linear attention, or sparse attention) introduces significant confounding variables which obscure the true source of performance gains. **Our work specifically targets the mathematical flaws of the Softmax attention mechanism itself**. Therefore, the only scientifically valid baseline is the standard Softmax attention equipped with identical RoPE, which allows us to isolate the contribution of our proposed mechanism.
>
> ### **2. Scaling Experiments on FineWeb-Edu**
>
> To address your request for validation across scales, we trained two new models on the **FineWeb-Edu (100B tokens)** dataset. The first is a 6-layer GPT-2-45M model with a configuration suggested by the Pythia suite for analysing scaling behaviours. The second is the standard GPT-2-355M architecture. Bothmodels were modified to incorporate RoPE to ensure a fair comparison
> with the state-of-the-art extrapolation baseline, and were trained with
> a sequence length of 1024 tokens.
>
>
> We specifically chose FineWeb-Edu over standard FineWeb. FineWeb-Edu is rigorously filtered using Llama-3-70B to retain only content with high educational value and logical coherence.
>
> Crucially, recent research highlights that training on such clean, high-quality, "textbook-grade" dataset drives models to learn robust reasoning patterns and focus sharply on semantic and syntactic coherence [1]. This characteristic actually **increases the difficulty of the Passkey Retrieval task**, where the model must attend to a high-entropy passkey token that acts as semantic noise within a highly coherent context. In this rigorous setting, the failure of Softmax becomes even more pronounced, while LSSAR demonstrates robustness.
>
> ### **3. Experimental Results**
>
> **A. Validation Loss Extrapolation**
>
> The standard Softmax attention exhibits a significant degradation in validation loss as the sequence length
> increases, failing to extrapolate effectively even when trained on high-quality data. In contrast, LSSAR maintains a stable and nearly constant validation loss across all tested lengths for both the 45M and 355M models, demonstrating that its entropy invariance property holds true regardless of
> model scale or data quality.
>
> *GPT-2-45M :*
> | Method | 1K | 2K | 4K | 8K | 16K |
> | :--- | :--- | :--- | :--- | :--- | :--- |
> | Softmax | 3.3269 | 4.2023 | 5.2937 | 5.9778 | 6.3841 |
> | **LSSAR** | **3.3407** | **3.3414** | **3.4494** | **3.5168** | **3.6942** |
>
> *GPT-2-355M :*
> | Method | 1K | 2K | 4K | 8K |
> | :--- | :--- | :--- | :--- | :--- |
> | Softmax | 2.6804 | 3.8234 | 5.4412 | 6.4913 |
> | **LSSAR** | **2.6850** | **2.6655** | **2.6223** | **2.7158** |
>
> **B. Downstream Tasks**
>
> LSSAR consistently outperforms Softmax. Notably, on **SummScreen** (Long-Context Summarization) with GPT-2 Medium, LSSAR achieves a **~4x improvement** over the baseline. These gains confirm that the architectural improvements of LSSAR translate into tangible benefits for complex reasoning and summarisation tasks, without sacrificing generic capabilities.
>
> *GPT-2-45M:*
> | Method | ARC-E | ARC-C | HellaSwag | PIQA | MMLU | SciQ | SummScreen |
> | :--- | :--- | :--- | :--- | :--- | :--- | :--- | :--- |
> | Softmax | 45.58 | 16.72 | 27.34 | 58.98 | 22.94 | 70.40 | 0.8100 |
> | **LSSAR** | **46.04** | **18.34** | **27.49** | **59.79** | **22.97** | **75.00** | **2.1932** |
>
> *GPT-2-355M:*
> | Method | ARC-E | ARC-C | HellaSwag | PIQA | MMLU | SciQ | SummScreen |
> | :--- | :--- | :--- | :--- | :--- | :--- | :--- | :--- |
> | Softmax | 59.30 | 23.46 | 30.63 | 66.49 | 22.98 | 78.90 | 2.4506 |
> | **LSSAR** | **62.50** | **26.96** | **34.81** | **68.34** | **23.94** | **84.00** | **9.5083** |

---

> > ### Author Response · Authors · 2025-12-01
> >
> > ### **Reply(2/2)**
> >
> > **C. Passkey Retrieval**
> >
> > Consistent with our hypothesis regarding high-quality training data, the standard Softmax
> > baseline fails completely, yielding 0% accuracy across all tested sequence lengths, including the
> > training window itself. The model’s strong bias towards coherent semantic structures prevents it
> > from attending to the random passkey, treating it effectively as noise. In stark contrast, LSSAR
> > successfully overcomes this limitation, achieving substantial retrieval accuracy within the training
> > length (57% for GPT-2-45M and 65% for GPT-2-355M) and maintaining functional capabilities into
> > the extrapolation regime. This result confirms that the proposed re-weighting mechanism effectively
> > sharpens the attention distribution, enabling the model to capture critical high-entropy information
> > even when trained on “textbook-quality” data that discourages such behaviour.
> >
> > *GPT-2-45M:*
> > | Method | 1K | 1.5K | 2K | 2.5K | 3K | 3.5K | 4K |
> > | :--- | :--- | :--- | :--- | :--- | :--- | :--- | :--- |
> > | Softmax | 0 | 0 | 0 | 0 | 0 | 0 | 0 |
> > | **LSSAR** | **57** | **16** | **3** | 0 | 0 | 0 | 0 |
> >
> > *GPT-2-355M:*
> > | Method | 1K | 1.5K | 2K | 2.5K | 3K | 3.5K | 4K |
> > | :--- | :--- | :--- | :--- | :--- | :--- | :--- | :--- |
> > | Softmax | 0 | 0 | 0 | 0 | 0 | 0 | 0 |
> > | **LSSAR** | **65** | **12** | 0 | 0 | 0 | 0 | 0 |
> >
> > These results strongly support that LSSAR provides a fundamental mechanical improvement over Softmax, regardless of model scale or data distribution.
> >
> > Best regards,
> >
> > The Authors
> >
> > ***
> >
> >
> > [1] Textbooks Are All You Need

---

### Official Review · Reviewer_8cFh · 2025-10-31

**Soundness:** 3
**Presentation:** 2
**Contribution:** 2
**Rating:** 4
**Confidence:** 3

**Summary:**

This paper introduces a Length Scaled Softplus Attention (LSSA) combined with an re-weighting scheme, aiming to replace the standard Softmax attention mechanism used in transformers. The proposed model (LSSAR) excels in length extrapolation while ensuring numerical stability, as demonstrated in the long-context passkey retrieval tasks.

**Strengths:**

1. The decomposition of Softmax operation into none-linear positive transformation and l1-norm is conceptually direct and simple to understand. The re-interpretation helps unify multiple Softmax-free attention variants under a coherent framework.

2. Supported by both quantitative results and visualization analyses, the proposed method shows stronger abilities in improving length extrapolation and reducing attention sink, suggesting that the two-stage normalization and re-weighting design offers a practical and effective improvement over conventional Softmax.

**Weaknesses:**

1. While the proposed method (LSSA/LSSAR) is conceptually solid and supported by several illustrative experiments, evaluation across a broader set of backbones is necessary to demonstrate robustness and effectiveness beyond a single model family. More baselines should be added to long-context tasks and downstream evaluation.

2. Since efficiency is central to practical deployment, the measurements of runtime or memory profiling relative to optimized kernels are critical for completeness.

**Questions:**

1. Could the authors provide a theoretical justification or mathematical analysis to support why normalization plays such a fundamental role in Softmax attention?

2. The paper states that N is an L×L matrix where each element in row i is equal to i, however, it remains unclear whether N is strictly lower-triangular (reflecting causal masking) or a dense matrix? Furthermore, how does logN behave for the first few tokens where Ni = 1?

3. Section 2.2 briefly mentions cosine similarity attention when discussing scaling factors, but the relationship between this mechanism and the proposed LSSA formulation remains unclear.

4. Why ReLU is chosen for the re-weighting mechanism instead of smoothing functions such as Sigmoid/GELU?

5. Considering that Softplus and Sigmoid are mathematically related through their derivatives, and the results in Table 4 show that the performance of Sigmoid Attention (Ramapuram et al., 2024) is consistently close to that of the proposed LSSAR, could the authors analyze the relationship between the two methods?

6. Can LSSAR be directly incorporated into efficient attention frameworks (e.g., FlashAttention, Mamba-2) without loss of parallelism?

---

> ### Author Response · Authors · 2025-11-18
>
> ### **Reply(1/3)**
>
> Dear 8cFh,
>
> We would like to sincerely thank you for your thorough and insightful review of our paper. Your questions are highly constructive and have prompted us to clarify several key aspects of our work, which we believe has significantly strengthened the manuscript. We have carefully considered all your points and provide our detailed responses below. We have also uploaded a revised version of the paper that incorporates these clarifications and improvements for your consideration.
>
> ### **Response to Weakness 1**
>
> We thank the reviewer for the constructive feedback regarding the generalizability and evaluation of our method. We agree that demonstrating robustness is paramount and address the specific points below.
>
> Our selection of a modified GPT-2 architecture was a principled methodological choice designed to create a controlled environment. Despite variations in scale and specific configurations, many modern LLMs share a core foundation: the Transformer architecture, Softmax attention, and RoPE. To ensure our findings are relevant to these modern models, we explicitly replaced the original absolute position embeddings in GPT-2 with RoPE for length extrapolation. This creates a controlled and representative testbed that mirrors the essential components of popular model families like Llama and Mistral, allowing for a focused analysis of the attention mechanism itself, which is the core contribution of our paper. Conducting extensive experiments from scratch on this standardized backbone provides clearer insights than fine-tuning various pre-trained models with their own inherent biases.
>
>
> We appreciate the suggestion to include more baselines. In our long-context evaluation (Table 4), we compared LSSAR not only against the standard Softmax but also against several state-of-the-art Softmax-free attention mechanisms, including recent Sigmoid-based and ReLU-based methods from prominent works (e.g., Ramapuram et al., 2024; Wortsman et al., 2023). We believe these represent the most relevant and direct competitors to our proposed attention mechanism. For the downstream evaluation, our goal was to demonstrate that the gains in length extrapolation translate to practical benefits, using the standard Softmax model as the primary and most critical baseline.
>
> We would also like to respectfully clarify that we have already addressed this potential limitation in **Section 4 (Discussion)** of our manuscript. In this section, we explicitly acknowledge that our evaluation was conducted on a single, small-scale model (GPT-2-124m) due to computational constraints. We then argue why the proposed LSSAR is expected to maintain its stability and performance when applied to larger models, **which directly relates to the reviewer's concern about robustness beyond a single model family**.
>
> ### **Response to Weakness 2**
>
> We appreciate the reviewer raising this critical point regarding practical efficiency.
>
> To provide a transparent and fair comparison, we have benchmarked the computational overhead of our proposed methods against the standard Softmax attention. Crucially, all implementations, including the baseline, are based on **standard PyTorch built-in functions** without leveraging specialized CUDA kernels like FlashAttention. This ensures a direct, apples-to-apples, comparison of the operations involved. The benchmark was run on an NVIDIA A100 GPU using bfloat16 precision (batch size=4, sequence length=1024). The results are presented below:
> | Configuration | Train Time (ms) | Train Memory (MB) | Eval Time (ms) | Eval Memory (MB) |
> | :--- | :--- | :--- | :--- | :--- |
> | Standard Attention | 120.48 | 9609.45 | 49.26 | 2324.71 |
> | LSSA | 169.92 | 12071.23 | 58.56 | 2325.58 |
> | LSSAR ($p=15$) | 250.32 | 16685.57 | 81.15 | 2325.58 |
>
> The results confirm that our current implementation introduces overhead. However, a closer look at the memory usage reveals a critical insight.
>
> Regarding the training memory, the significant increase in training memory for LSSAR ($p=15$) is an artifact of the PyTorch Autograd engine when handling the $\text{ReLU}(\cdot)^{p}$ operation. To compute gradients for this function in the backward pass, PyTorch must store the large intermediate tensor.  Specifically, the output of the ReLU function for every layer. As $p$ increases, the complexity of the gradient calculation ($p \cdot \text{ReLU}(\cdot)^{p-1}$) grows, and these cached activations accumulate rapidly in global memory.
>
> Crucially, please observe the Eval Memory. During inference (where gradients are not required), the memory footprint of LSSAR ($p=15$) is **identical** to that of Standard Attention (~2325 MB). This proves that the **model state itself is not larger**, and the intrinsic memory requirement of the algorithm is efficient.

---

> ### Author Response · Authors · 2025-11-18
>
> ### **Reply(2/3)**
> This stark contrast between Training and Eval memory demonstrates the massive optimization potential of a **Custom Fused Kernel**. A fused kernel (similar to FlashAttention) would compute the sharpening operations on the fly in fast SRAM, eliminating the need to materialize and store these massive intermediate tensors in HBM for backpropagation. This would bring the training memory consumption of LSSAR down to a level comparable to the optimized baseline.
>
> We believe this is a valuable addition that significantly strengthens the paper. **To fully address your concern, we have already incorporated these results and this detailed analysis into a new subsection titled "Computational Analysis" in the main body of the paper. The revised manuscript has now been uploaded**. We invite the reviewer to examine the new section. We hope this immediate action and the added transparency effectively resolve your concerns.
>
> ### **Response to Question 1**
>
> This is an excellent question. Our argument, presented in the **final paragraph of Section 3.1** of the original manuscript, is rooted in viewing the attention mechanism as a dynamic linear transformation of the value vectors ($\mathbf{V}$). The role of the $l_1$-norm is to constrain the rows of this transformation matrix.
>
> Specifically, by ensuring the weights in each row sum to one, the $l_1$-norm forces the output for each query to be a **convex combination** of the value vectors. A convex combination guarantees that the output vector will always lie within the convex hull of the input value vectors. This provides a strong inductive bias that prevents the output's magnitude from exploding, which is a critical factor for ensuring stable training dynamics in deep networks.
>
> ### **Response to Question 2**
>
> Thank you for this question, which helps us clarify an important detail.
>
> 1.  **Structure of $\mathbf{N}$:** In our formulation, $\mathbf{N}$ is a dense matrix where all elements in row $i$ have the value $i$, as defined in line 158.
>
> 2.  **Behavior at $i=1$:** For the first row ($i=1$), the scaling term $\log(\mathbf{N}_i)$ correctly evaluates to $\log(1) = 0$, and subsequently Softplus(0) yields $\log(2)$. This outcome is an intentional and stable property of the formulation. Conceptually, this applies to the first token, which can only attend to itself. In any attention scenario with a single query-key pair, the post-normalization score is definitionally 1. Consequently, while a scalar value is applied, it does not alter the final attention weight in this specific context. Our mechanism robustly handles this boundary condition without any instability.
>
> ### **Response to Question 3**
>
> This is a great question that gets to the heart of our design motivation. The relationship between our work and cosine similarity is **intrinsic and structural**. LSSA is fundamentally grounded in the cosine similarity mechanism, and the discussion in Section 2.2 serves as the **theoretical imperative for our proposed scaling factor**.
>
> The logic is as follows:
> 1.  Our LSSA formulation (Eq. 4) begins by applying $l_2$-normalization to the query and key vectors, which transforms the dot product into a cosine similarity calculation.
> 2.  However, as we discuss, standard cosine similarity is susceptible to a vanishing gradient phenomenon in high-dimensional spaces, where vectors tend toward orthogonality, causing their dot products to converge to zero.
> 3.  Since LSSA adopts this cosine formulation, it inherently inherits this risk. This makes the introduction of a scaling mechanism an **essential correction**, not an optional heuristic.
>
> This necessity is precisely what motivates our design of the dynamic scaling factor $(\log d \log \mathbf{N})$. It is specifically engineered to counteract the signal decay caused by high dimensionality ($d$) and sequence length ($\mathbf{N}$), ensuring that attention scores remain within an effective dynamic range for the Softplus activation.

---

> ### Author Response · Authors · 2025-11-18
>
> ### **Reply(3/3)**
>
> ### **Response to Question 4**
>
> The decision to use ReLU was a deliberate one, directly motivated by the core challenge of attention smoothing in long sequences. Our re-weighting mechanism is based on the assumption that for any given token, many tokens appearing much earlier in the sequence are semantically irrelevant. Consequently, an effective attention mechanism should be able to **completely disregard** these irrelevant tokens rather than assigning them small, non-zero weights.
>
> ReLU is uniquely suited for this purpose due to its inherent **sparsity**. By centering the attention scores around zero and then applying ReLU, we effectively perform a hard masking operation, setting the scores of presumably irrelevant tokens to exactly zero. This aligns perfectly with our initial assumption. In contrast, smoother functions like Sigmoid or GELU would assign small, positive weights to all tokens, preventing the model from fully ignoring contextual noise. This initial sparsity introduced by ReLU also significantly enhances the efficiency of the subsequent power-law sharpening step ($p$), as it can more effectively amplify the truly significant scores.
>
> ### **Response to Question 5**
>
> The reviewer's observation is astute. The performance of the modified Sigmoid attention is indeed close to our LSSA, but we argue this similarity does not stem from a deep mathematical connection between the two activation functions in this context. Instead, it highlights the critical impact of the two architectural principles we introduce: the dynamic length scale factor ($\log \mathbf{N}$) and the explicit $l_1$-norm.
>
> As shown in Table 4, the original Sigmoid attention from Ramapuram et al. (2024) without these modifications performs poorly, especially on longer sequences. The performance becomes competitive **only when we augment it with our proposed $\log \mathbf{N}$ scaling and $l_1$-norm**. This strongly suggests that the high performance of both methods is primarily attributable to this shared design philosophy, rather than the choice between Sigmoid and Softplus at the normalization stage.
>
> Furthermore, the distinction becomes clearer when considering our full LSSAR model, which includes the re-weighting stage. As the results for $p=15$ in Table 4 demonstrate, our Softplus-based LSSAR maintains superior performance and stability under the sharpening effect of re-weighting. This indicates that while both functions can be effective within our proposed normalization framework, Softplus provides a more robust foundation for the subsequent sharpening stage.
>
> ### **Response to Question 6**
>
> We thank the reviewer for this crucial question regarding practical implementation. It is correct to distinguish between these frameworks.
>
> LSSAR is designed as a direct replacement for the Softmax attention mechanism within the Transformer architectural paradigm. Consequently, it is not compatible with State Space Models like Mamba-2, which eschew the quadratic attention mechanism altogether.
>
> LSSAR is **highly amenable** to optimizations like FlashAttention. The core operations of LSSAR, including element-wise scaling, Softplus, the power function, and the final row-wise normalization, are all local and do not introduce sequential dependencies that would hinder the block-wise, parallel computation central to FlashAttention's I/O-aware algorithm. The performance overhead observed in our experiments is an artifact of our proof-of-concept implementation using a sequence of standard, non-fused PyTorch operations. As we noted in our discussion in Section 4, we are confident that a fused CUDA kernel could be developed for LSSAR to achieve computational efficiency on par with highly optimized attention implementations.

---

### Author Response · Authors · 2025-11-26

Dear Reviewer,

I hope this message finds you well.

With the discussion period ending in **less than seven days**, I wanted to follow up and ensure that our rebuttal and the revisions to our manuscript have satisfactorily addressed your concerns.

Your insights have been invaluable to us. If there are any additional points or further feedback you would like us to consider, please do not hesitate to let us know. We are eager to address any remaining issues.

Thank you for your time and for your thorough review of our paper.

Sincerely,

The Authors

---

### Meta-Review · Area_Chair_XgRn · 2026-01-10

**Summary:**

This paper proposes a novel attention mechanism to address numerical instability and length extrapolation limitations in standard Softmax attention. By decomposing the Softmax operation, the authors identify the l1-norm as the critical component. Consequently, this paper replaces the exponential operation with a Softplus function and further introduces a re-weighting mechanism that sharpens the attention distribution via a shifted ReLU-p transformation. The method introduced by this paper is sound and well-motivated. It has demonstrated advantages over Softmax in length extrapolation, passkey retrieval, and the mitigation of the attention sink phenomenon under some experimental scenarios. However, it suffers from insufficient experiments. The reviews raised concerns regarding weak baselines, with the primary concern being the toy-scale evaluation and limited architectures. The authors only evaluated on the GPT-2 family with relatively small scales, which is too limited to support strong conclusions in the context of the modern LLM landscape. I believe this paper provides a meaningful perspective; however, experimental sufficiency regarding practicality and generalization is also critical. Therefore, my recommendation is rejection.

**Reviewer Concerns:**

Scalability: Although the authors have added experiments with two additional model sizes, these remain within the "toy" scale range and may not adequately address reviewer concerns regarding scalability.

Computational analysis: The authors have provided new measurements for memory usage and runtime. However, under the native PyTorch implementation, both training and inference times for LSSAR are slower than standard Attention. This may raise further concerns regarding the practical utility of the method.

Outstanding Issues:

- The fundamental defect of relying on "toy-scale" experimental evaluations.

- The lack of comparison against strong modern baselines.

- Challenges regarding practical engineering deployment.

**Reviewer Scores:**

The reviewers may maintain their scores.

---

### Decision · Program_Chairs · 2026-01-26

Reject